# Schedule Your Edit: A Simple yet Effective Diffusion Noise Schedule for Image Editing

**Haonan Lin[1]**   **Yan Chen[1]**   **Jiahao Wang[1]**   **Wenbin An[3]**   **Mengmeng Wang[2,5]***

**Feng Tian[1]**   **Yong Liu[4,5]**   **Guang Dai[5]**   **Jingdong Wang[6]**   **Qianying Wang[7]**

[1] School of Comp. Science & Technology, MOEKLINNS Lab, Xi'an Jiaotong University
[2] College of Comp. Science & Technology, Zhejiang University of Technology
[3] School of Auto. Science & Engineering, MOEKLINNS Lab, Xi'an Jiaotong University
[4] Institute of Cyber-Systems and Control, Zhejiang University
[5] SGIT AI Lab, State Grid Corporation of China
[6] Baidu Inc   [7] Lenovo Research

## Abstract

Text-guided diffusion models have significantly advanced image editing, enabling high-quality and diverse modifications driven by text prompts. However, effective editing requires inverting the source image into a latent space, a process often hindered by prediction errors inherent in DDIM inversion. These errors accumulate during the diffusion process, resulting in inferior content preservation and edit fidelity, especially with conditional inputs. We address these challenges by investigating the primary contributors to error accumulation in DDIM inversion and identify the singularity problem in traditional noise schedules as a key issue. To resolve this, we introduce the *Logistic Schedule*, a novel noise schedule designed to eliminate singularities, improve inversion stability, and provide a better noise space for image editing. This schedule reduces noise prediction errors, enabling more faithful editing that preserves the original content of the source image. Our approach requires no additional retraining and is compatible with various existing editing methods. Experiments across eight editing tasks demonstrate the *Logistic Schedule*'s superior performance in content preservation and edit fidelity compared to traditional noise schedules, highlighting its adaptability and effectiveness. (Project page: `https://lonelvino.github.io/SYE/`)

## 1   Introduction

Text-guided diffusion models have emerged as a leading technique in image generation, offering remarkable visual quality and diversity [2, 42, 50, 69]  . The noise latent space of these models can be leveraged to retain and modify images [32, 66, 68], enabling text-guided editing where a source image is adjusted based on a target prompt. This requires first inverting the source image into a latent variable (*e.g.*, via DDIM inversion), due to the absence of its predefined latent space [28, 39].

While DDIM inversion proves effective for unconditional diffusion models [43, 55], it results in inferior content preservation and suboptimal edit fidelity when applied to conditional inputs [12, 17]. This phenomenon is particularly evident in image editing, which requires incorporating new conditionals into the generation process [16, 59, 33, 61]. DDIM converts the DDPM into a

---

*Corresponding Author.
This work was completed during the internship at SGIT AI Lab, State Grid Corporation of China.

38th Conference on Neural Information Processing Systems (NeurIPS 2024).

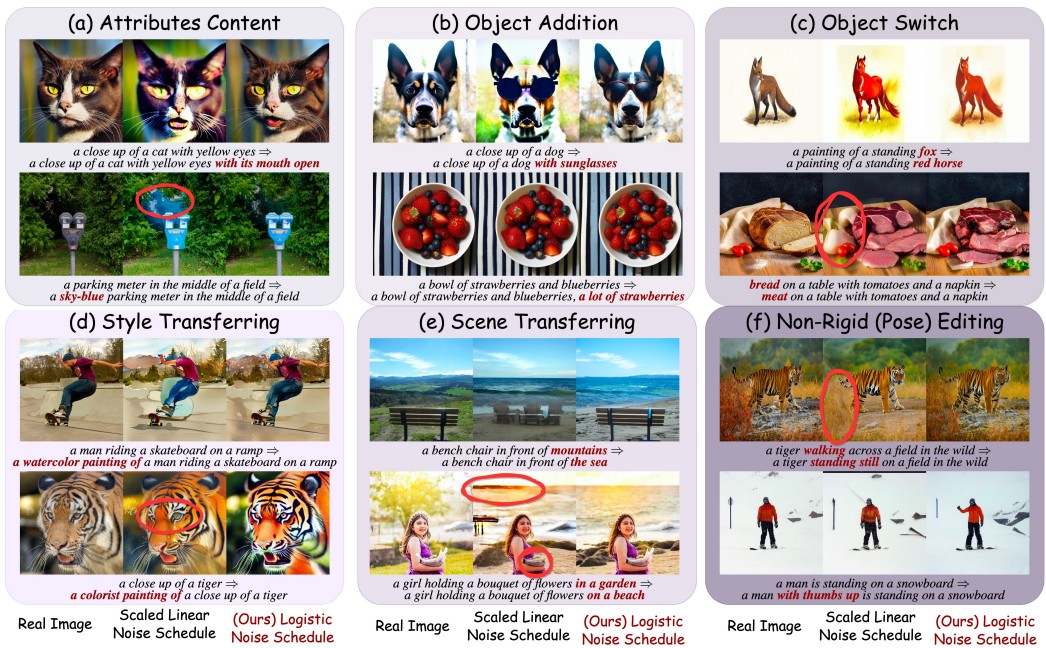

Figure 1: Compared to linear noise schedule, *Logistic Schedule* ❶ demonstrates high fidelity in attributes content editing (a, b) with EF-DDPM [21], ❷ preserves the high-level semantics of the source image while performing object translation (c) with pix2pix-zero [45] and style/scene transferring (d, e) with StyleDiffusion [63], and ❸ successfully conducts non-rigid alteration (f) via MasaCtrl [6]. Text prompts corresponding to each input image are presented beneath each sample, with words introduced for image editing distinctly highlighted in red.

deterministic process by approximating the Markov process as a non-Markov process based on a local linearization assumption [55]. This approximation introduces noise prediction errors that accumulate throughout the diffusion process, leading to deviations in the inverted latent representation from its original distribution, as illustrated in Fig. 2 left. Recently, inversion-based editing methods have emerged as a promising paradigm to address these issues by aligning the reconstruction path more closely with the DDIM inversion trajectory, thereby ensuring the preservation of the original content in the edited images [41, 15, 44, 10, 25]. However, these methods still heavily rely on the accuracy of the DDIM inversion. This leads us to a fundamental question: **What if we correct the DDIM inversion errors to naturally reduce the loss of original content in the edited images?**

Unlike previous inversion-based editing methods that focus on minimizing the distance between $\mathbf{x}_t''$ and $\mathbf{x}_t^*$ (Fig. 2 left), we investigate the primary reason for error accumulation in DDIM inversion. Based on the fact that DDIM samplers can be derived by deterministic ODE processes [3, 38, 71], our analysis reveals that these traditional noise schedule designs result in a singularity problem (Fig. 2 right) when treating the DDIM inversion process as solving a differentiable ODE. This results in unreliable noise predictions from the start, and as errors accumulate, the editing results degrade (Fig. 1). This insight motivates us to address the problem at its source: the noise schedule itself. To our knowledge, this is the first work focusing on designing noise schedules specifically for image editing, providing an optimized solution without requiring complete model retraining [14, 20, 23, 29, 26, 34].

We present a simple yet effective noise schedule, *Logistic Schedule*, designed to resolve the singularity problem of previous noise schedules and enhance inverted latents for image editing. The key ideas behind *Logistic Schedule* are twofold: (1) creating a well-defined noise schedule to improve inversion stability, and (2) providing a better noise space that enables editing faithful to the source image. Specifically, *Logistic Schedule* eliminates singularities at the beginning of the inversion process, thereby reducing noise prediction errors in the inverted latents. It enables more stable data perturbation to preserve the original content of the source image in the edited image. Importantly, this design is effective and compatible with other editing methods without requiring additional retraining.

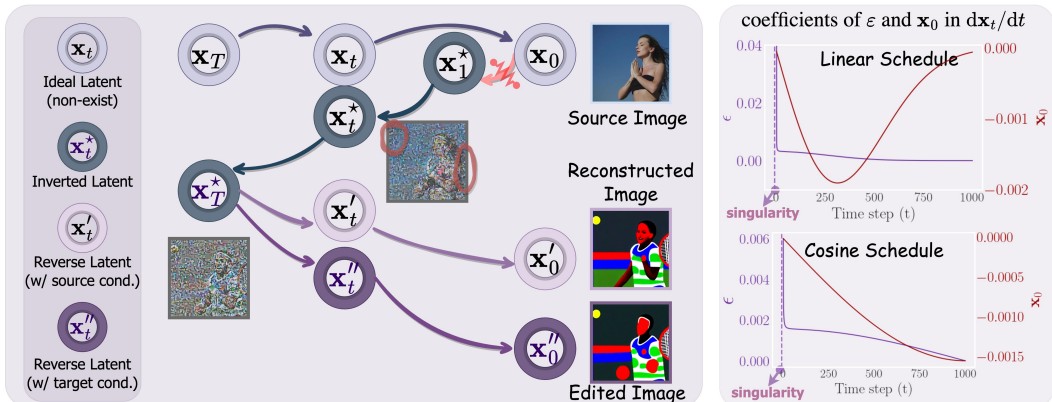

Figure 2: **Illustration of the DDIM inversion in image editing and its challenges**. Left: starting from the source image $\mathbf{x}_0$, the ideal latent $\mathbf{x}_t$ is approximated by the inverted latent $\mathbf{x}_t^*$ using DDIM inversion. The perturbed noisy latent $\mathbf{x}_T^*$ is then sampled in two branches—one for the source condition and one for the target condition—yielding the reconstructed and edited images respectively. Right: the numerical computations of $\mathrm{d}\mathbf{x}_t/\mathrm{d}t$ for scaled linear and cosine noise schedules, highlighting the singularity at $t = 0$ that leads to potential inaccuracies in noise prediction during inversion.

We conducted experiments across eight distinct editing tasks using approximately 1600 images from diverse scenes. Fig. 1 illustrates that our *Logistic Schedule* effectively enhances editing results in terms of essential content preservation and edit fidelity compared to commonly used noise schedules like the linear schedule. Moreover, our schedule can be seamlessly integrated with various existing diffusion-based editing techniques, demonstrating its versatility and effectiveness. Our main contributions are summarized as follows: (1) **Theoretical Analysis**: We analyze the failure of DDIM inversion in real-image editing step by step, identifying the singularity in the noise schedule as the key issue to address. (2) **Methodology**: We introduce *Logistic Schedule*, a novel diffusion noise schedule specifically tailored for real-image editing, which effectively reduces prediction errors during inversion. (3) **Superiority**: We showcase *Logistic Schedule*'s adaptability by integrating it with various editing methods and demonstrate its consistent superior performance across different editing tasks.

## 2   Background

This section will introduce diffusion models and their noise schedules, along with DDIM inversion, which are crucial for text-guided editing of real images.

**Diffusion Models.** Denoising Diffusion Probabilistic Models (DDPM) [18] are designed to transform a random noise vector $\mathbf{x}_T$ into a series of intermediate samples $\mathbf{x}_t$, and eventually a final image $\mathbf{x}_0$ by progressively adding Gaussian noise $\epsilon \sim \mathcal{N}(0, \mathbf{I})$ according to a noise schedule $\beta_1, \ldots, \beta_T$:

$$\mathbf{x}_t = \sqrt{1 - \beta_t}\mathbf{x}_{t-1} + \sqrt{\beta_t}\epsilon_{t-1},$$

where $t \sim [1, T]$ and $T$ denotes the number of timesteps. The noise schedule determines the distribution of noise scales and is designed to ensure that the noise scale at each step is proportional to the remaining signal, which is usually fixed without additional learning. According to the properties of conditional Gaussian distributions, $\mathbf{x}_t$ can be derived from a real image $\mathbf{x}_0$ in the following closed form by reparameterizing $\alpha_t = 1 - \beta_t$, $\bar{\alpha}_t = \prod_{i=1}^{t} \alpha_i$:

$$\mathbf{x}_t = \sqrt{\bar{\alpha}_t}\mathbf{x}_0 + \sqrt{1 - \bar{\alpha}_t}\epsilon. \tag{1}$$

Another commonly used sampling method is Denoising Diffusion Implicit Models (DDIM) [55], which formulate a denoising process to generate $\mathbf{x}_{t-1}$ from a sample $\mathbf{x}_t$ via:

$$\mathbf{x}_{t-1} = \sqrt{\alpha_{t-1}}\underbrace{\left(\frac{\mathbf{x}_t - \sqrt{1 - \alpha_t}\epsilon_\theta^{(t)}(\mathbf{x}_t)}{\sqrt{\alpha_t}}\right)}_{\text{predicted } \mathbf{x}_0} + \underbrace{\sqrt{1 - \alpha_{t-1} - \sigma_t^2} \cdot \epsilon_\theta^{(t)}(\mathbf{x}_t)}_{\text{direction pointing to } \mathbf{x}_t} + \underbrace{\sigma_t \epsilon_t}_{\text{random noise}}, \tag{2}$$

where $\epsilon_t \sim \mathcal{N}(0, \mathbf{I})$, $\sigma_t$ is the variance schedule, and $\epsilon_\theta$ is a network trained to predict the noise added. When $\sigma_t = \sqrt{\frac{1-\alpha_{t-1}}{1-\alpha_t}}\sqrt{1 - \frac{\alpha_t}{\alpha_{t-1}}}$ for all $t$, the forward process becomes Markovian, and the generation process becomes a DDPM. And in a special case when $\sigma_t = 0$ for all $t$, the forward process become deterministic given $\mathbf{x}_{t-1}$ and $\mathbf{x}_0$, except for $t = 1$, and the generate process becomes a DDIM.

**Inversion in Image Editing.** Although text-to-image diffusion models [50, 52, 19] have advanced feature spaces that support various downstream tasks [67, 37, 36], applying them to real images (non-generated images) is challenging because these images lack a natural diffusion feature space. Editing a real image first requires obtaining the latent variables $\mathbf{x}_T$ from the original image $\mathbf{x}_0$ and then performing the generation process under new conditions. To bridge this gap, DDIM inversion [55] is predominantly used due to its deterministic process, which can be represented by reversing the generation process in Eq. 2 with $\sigma_t = 0$:

$$\mathbf{x}_t^* = \frac{\sqrt{\alpha_t}}{\sqrt{\alpha_{t-1}}}\mathbf{x}_{t-1}^* + \sqrt{\alpha_t}\left(\sqrt{\frac{1}{\alpha_t} - 1} - \sqrt{\frac{1}{\alpha_{t-1}} - 1}\right)\epsilon_\theta\left(\mathbf{x}_{t-1}^*, t-1\right).$$

However, existing editing methods that rely on vanilla DDIM inversion struggle to achieve both content preservation and edit fidelity when applied to real images [1, 4, 16]. Recently, inversion-based editing methods have improved the edited results by maintaining two simultaneous procedures: reconstruction and editing, as shown in Fig. 2 left. These methods align the reconstruction path ($\mathbf{x}'$) more closely with the DDIM inversion trajectory ($\mathbf{x}^*$), ensuring better preservation of the original content in the edited image [41, 15, 44, 25, 10]. Despite their effectiveness, these methods still heavily rely on the accuracy of the inverted latents obtained from DDIM inversion. In contrast, we start from a different perspective, focusing on improving the DDIM inversion accuracy to naturally enhance the edited results. In the following section, we begin with the transition from DDPM to DDIM, emphasizing the need for a better noise schedule for the inversion process.

## 3 On the Failure of DDIM Inversion

### 3.1 Warmup: Error Accumulation of DDIM

DDIM inversion for real images is unstable due to its reliance on a local linearization assumption at each step, leading to error accumulation and content loss from the original image. Specifically, DDIM assumes that the denoising process in Eq. 2 is roughly invertible, meaning $\mathbf{x}_t^*$ can be approximately recovered from $\mathbf{x}_{t-1}^*$ via:

$$\mathbf{x}_t^* = \frac{\mathbf{x}_{t-1}^* - b_t\epsilon(\mathbf{x}_t^*, t)}{a_t} \approx \frac{\mathbf{x}_{t-1}^* - b_t\epsilon(\mathbf{x}_{t-1}^*, t)}{a_t}, \tag{3}$$

where $a_t = \sqrt{\alpha_{t-1}/\alpha_t}$ and $b_t = -\sqrt{\alpha_{t-1}(1-\alpha_t)/\alpha_t} + \sqrt{1-\alpha_{t-1}}$. This approximation assumes $\epsilon(\mathbf{x}_t^*, t) \approx \epsilon(\mathbf{x}_{t-1}^*, t)$, and the inversion's accuracy depends on this assumption. However, ensuring accurate inversion under this assumption requires a sufficient number of discretization steps, which increases time costs and is impractical for many applications. With fewer timesteps or higher noise levels, error accumulation becomes more pronounced, resulting in distorted reconstructions, as shown in Fig. 2 left. This occurs because once we deviate from the linearization assumption, the interpolation operation in Eq. 3 fails. The primary issue arises when estimating the "predicted $\mathbf{x}_0$" in Eq. 2 at the initial step ($t = 1$, indicated by the red arrow in Fig. 2 left), where a simple expression for the posterior mean conditioned on $\mathbf{x}_t$ no longer exists [55]. Moreover, this problem is exacerbated in image editing, where the denoising process must incorporate new conditions into the image content. This increases the difficulty of noise predictions, leading to more severe artifacts and distortions.

### 3.2 The Devil Is in the Singularities

To get around this issue, our first insight is to reduce the prediction error at the beginning of the forward (inversion) process. But before we can figure out how to fix the error, we need to pinpoint the problem. We first provide the continuous generalization of DDPM, since sampling from diffusion models can be viewed alternatively as solving the corresponding ODE process [57, 38]:

$$d\mathbf{x} = \left[\mathbf{f}(\mathbf{x}, t) - \frac{1}{2}g(t)^2 \nabla_\mathbf{x} \log p_t(\mathbf{x})\right] dt, \tag{4}$$

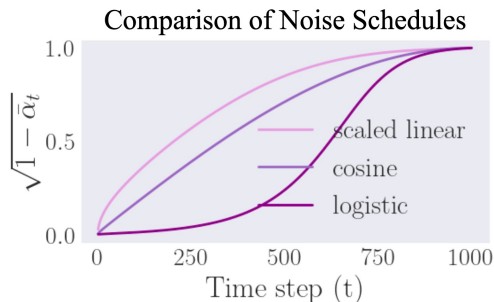 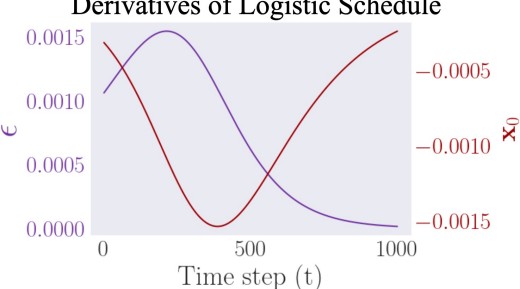

Figure 3: Left: trends of $\sqrt{1 - \alpha_t}$ (noise scales) for scaled linear, cosine, and logistic noise schedules. Right: $d\mathbf{x}_t / dt$ for the logistic schedule, highlighting its smooth transition, which prevents singularities and maintains the integrity of the initial latent vector $\mathbf{x}_0$.

where $\mathbf{f}(\cdot, t)$ is a vector-valued function called the *drift* coefficient of $\mathbf{x}(t)$, and $g(\cdot)$ is a scalar function known as the *diffusion* coefficient of $\mathbf{x}(t)$. And the ODE form of DDIM is equivalent to a special case of Eq. 4, as long as $\alpha_t$ and $\alpha_{t-\Delta t}$ are close enough (refer to details in *Appendix* A).

By treating the DDIM inversion process as solving a differentiable ODE, we emphasize that precise and stable computation of $d\mathbf{x}_t / dt$ at each timestep $t$ is crucial for accurate noise prediction, especially at the start of the inversion process. Fig. 2 right highlights the pitfalls of widely-used scaled linear [18] and cosine [43] noise schedules through numerical computations of $d\mathbf{x}_t / dt$.

**Proposition 3.1** (Singularity in Inversion Process). *During the inversion process, there exists a singularity at $t = 0$ for both the scaled linear and cosine schedule (Fig. 2 right):*

$$When \ t = 0, \ \left. \frac{d\mathbf{x}_t}{dt} \right|_{t \to 0} = \frac{0}{0} \cdot sign(\epsilon) = \infty \cdot sign(\epsilon).$$

*This singularity significantly affects the starting point of the inversion process during image editing tasks. Properly modeling $d\mathbf{x}_t / dt$ ensures that the inversion closely aligns with the true continuous dynamics of the diffusion process, thereby reducing errors and enhancing the fidelity of the inverted latents, which is critical for high-quality image editing. The proof can be found in Appendix B.*

We argue that singularities in modeling $d\mathbf{x}_t / dt$ cause significant issues in inversion-based text-guided image editing. Specifically, **(1) the instability in the inversion process** arises from the singularity of the derivatives at $t = 0$, leading to inaccurate noise component estimates, making the starting point inconsistent with the data's true characteristics. The fast sampling in DDIM exacerbates error accumulation, where minor initial errors lead to substantial deviations in the final inverted latents. As a result, reconstructed or edited images may display visual inconsistencies, distorted details, or unnatural artifacts, reducing the overall quality and fidelity. Furthermore, the singularity can also lead to **(2) poor handling of complex data distributions** in the real world. Discontinuities in derivatives result in the model receiving inconsistent and unreliable signals during the diffusion probabilistic modeling. This hinders the model's ability to capture intricate patterns and details, disrupting the consistency and integrity within an image [30].

## 4  Better Noise Schedule Helps Inversion and Editing

### 4.1  Well-Defined Schedule Improve Inversion Stability

To address the issues highlighted in *Proposition* 3.1, we propose a new noise schedule in terms of $\bar{\alpha}_t$, since $\bar{\alpha}_t$ represents the remaining signals in the latents during the diffusion process (Eq. 1). Following the recommendations from iDDPM [43], the noise schedule should ensure that noise is added more slowly at the beginning to preserve image information in the middle of the diffusion process. We introduce our logistic noise schedule as follows:

$$\bar{\alpha}_t = \frac{1}{1 + e^{-k(t-t_0)}}, \tag{5}$$

where $k$ and $t$ are hyperparameters that control the steepness and midpoint of the logistic function, respectively. In our experiments, we set $k = 0.015$ and $t_0 = \text{int}(0.6T)$, as discussed in Section 5.3.1.

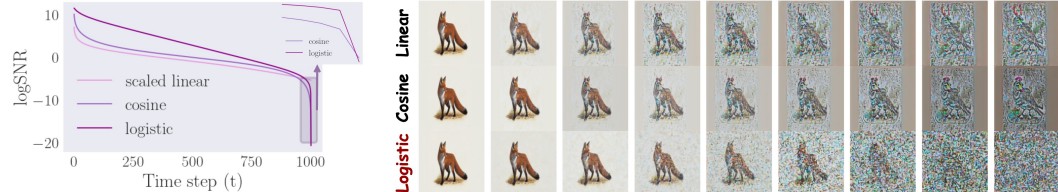

Figure 4: **Analysis of noise space for different schedules**. Left: logSNR trends, where the logistic schedule maintains a more gradual decline. Right: inversion processes, with the logistic schedule preserving more details in the initial stage and minimizing low-frequency retention in the final stage.

Our logistic schedule is designed to have a linear drop-off of $\alpha_t$ in the middle of the diffusion process, with minimal changes near the extremes of $t = 0$ and $t = T$, thus preventing abrupt changes in the noise level. Fig. 3 left demonstrates the progression of $\sqrt{1 - \alpha_t}$ for different schedules, in which linear and cosine schedules tend to add noise too quickly during the early stage of the inversion process. Crucially, our logistic noise schedule avoid the singularity of $\mathrm{d}\mathbf{x}_t/\mathrm{d}t$ at $t = 0$. For simplicity in expression, we set $k = 0.015$ and $t_0 = 30$, resulting in the following:

$$\text{When } t = 0, \left. \frac{\mathrm{d}\mathbf{x}_t}{\mathrm{d}t} \right|_{t \to 0} = 1.486e^{-3}\epsilon - 1.318e^{-3}\mathbf{x}_0$$

The proof is provided in *Appendix* B, and the trend of the derivatives of our logistic schedule is illustrated in Fig. 5 right. By ensuring a smooth and continuous transition in noise levels, the logistic schedule maintains the integrity of the initial latent vector $\mathbf{x}_0$. This alignment with the diffusion process's continuous dynamics prevents undesired deviations, reduces errors, and leads to more accurate and stable latent predictions, improving the inversion process's fidelity.

### 4.2 Exploring Noise Space of Logistic Schedule

We now explore the properties of our logistic noise schedule and its influence on the noise space, specifically comparing the logSNR trends and inversion processes of different noise schedules.

**Steady Information Perturbation.** As depicted in Fig. 4 (left), the linear and cosine schedules tend to drastically degrade image information at the initial stage of inversion, as evidenced by the rapid drop in logSNR. In contrast, our logistic schedule exhibits a more linear decrease in logSNR before the final stage, ensuring steady data perturbation. This steadiness allows the logistic schedule to capture a richer set of features and nuances from the original image, facilitating more detailed reproduction and higher fidelity in the edited images.

**Comprehensive Pattern Capture.** As shown in Fig. 4 (right), we visualize the latents during the inversion (forward) process, using 50 timesteps with the final step at 981 instead of 999. In the early stage, our *Logistic Schedule* preserves more original image information, reflecting the logSNR trend. Considering the later stage, linear and cosine schedules retain more low-frequency components due to higher endpoint SNRs, explaining why their noise maps don't fully cover the image. In contrast, our *Logistic Schedule* ensures that the inverted latent closely resembles pure Gaussian noise, minimizing the retention of low-frequency components. This thorough process ensures that the inversion encodes a broader array of the original image's information, thereby enhancing the quality and fidelity of the edited images.

## 5 Experiments

In the section below, we evaluate our method both quantitatively and qualitatively on text-guided editing of real images. To validate the versatility and effectiveness of our proposed *Logistic Schedule*, we compare it with linear and concise schedules by employing different editing approaches across various editing tasks. Refer to *Appendix* E for detailed experimental results.

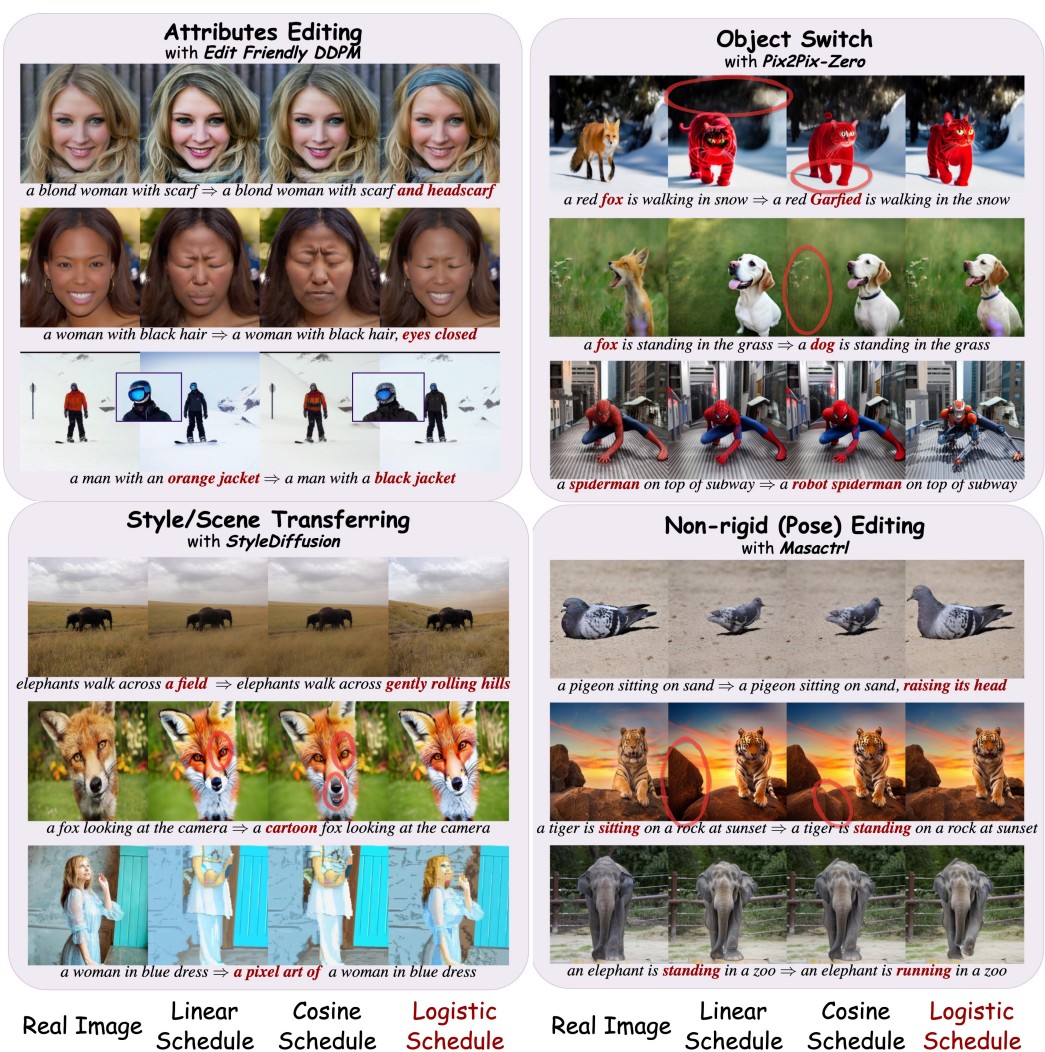

Figure 5: **Qualitative comparison of the *Logistic Schedule* with linear and cosine schedules across various image editing tasks**. To preserve background content during ① attribute editing tasks (*e.g.*, colors, and materials), we employ *Edit Friendly DDPM* [21]; for tasks requiring background preservation such as ② object translation, we use *Zero-shot Pix2Pix* [45]; for tasks involving ③ scene or style transfer, while maintaining object semantics, we utilize *StyleDiffusion* [63]; to validate spatial context preservation in ④ non-rigid editing tasks (*e.g.*, motion, pose), we consider *MasaCtrl* [6].

## 5.1 Experimental Settings

**Implementation Details.** We perform the inference of different editing and inversion methods under consistent conditions. We use Stable Diffusion v1.5 as the base model, with 100 timesteps, an inversion guidance scale of 3.5, and a reverse guidance scale of 7.5. All experiments are conducted on a single Nvidia A100 GPU. Quantitative results are averaged over 10 random runs. Additional implementation details are provided in *Appendix* D.2.

**Datasets.** Experiments are conducted on the PIEBench dataset [25]. Recognizing the dataset's limited size and scenarios, we extend it by incorporating face images from FFHQ [27] and AFHQ [11], as well as indoor/outdoor common objects from MS-COCO [35]. This results in approximately 1600 images in total, across eight editing types (see *Appendix* D.1).

**Evaluation Metrics.** As the editing process involves altering both the foreground and background of the images, we follow Ju et al. in adopting three types of metrics: structure (DINO-I [7, 58, 51]),

background preservation (PSNR, LPIPS [24, 72], MSE, SSIM [64]), and image-image, text-image consistency (CLIP score [48]). Detailed descriptions of each metric can be found in *Appendix* D.3.

Table 1: Comparative table of diffusion noise schedules and their performance metrics. Bold values indicate the best results, while underlined values denote the second-best results.

| Schedule | Structure Dist $_{\times 10^{-3}}$ ↓ | PSNR ↑ | Background Preservation | | | CLIP Similarity (%) | |
| | | | LPIPS $_{\times 10^{-3}}$ ↓ | MSE $_{\times 10^{-4}}$ ↓ | SSIM $_{\times 10^{-2}}$ ↑ | Visual ↑ | Textual ↑ |
|---|---|---|---|---|---|---|---|
| **Attributes Editing (with *Edit Friendly DDPM*)** | | | | | | | |
| Linear | 35.66 | 20.70 | 134.88 | 113.61 | 77.60 | 79.82 | 23.06 |
| Cosine | 26.57 | 22.38 | 110.52 | 80.01 | 80.15 | 81.35 | 22.39 |
| **Logistic** | **17.37** $_{34.6\%↓}$ | **24.78** $_{10.7\%↑}$ | **81.80** $_{26.0\%↓}$ | **49.47** $_{38.2\%↓}$ | **82.97** $_{3.5\%↑}$ | **82.44** $_{0.8\%↑}$ | **23.62** $_{2.4\%↑}$ |
| **Object Switch (with *Zero-Shot Pix2Pix*)** | | | | | | | |
| Linear | 39.02 | 19.93 | 134.64 | 138.99 | 74.63 | 83.33 | 22.30 |
| Cosine | 30.83 | 21.15 | 113.03 | 107.46 | 77.23 | 84.32 | 22.46 |
| **Logistic** | **22.4** $_{27\%↓}$ | **22.91** $_{8\%↑}$ | **90.75** $_{20\%↓}$ | **82.05** $_{24\%↓}$ | **79.32** $_{3\%↑}$ | **84.52** $_{0.1\%↑}$ | **22.65** $_{0.8\%↑}$ |
| **Style/Scene Transferring (with *StyleDiffusion*)** | | | | | | | |
| Linear | 38.06 | 21.17 | 93.70 | 111.01 | 81.85 | 77.65 | **25.39** |
| Cosine | 28.44 | 22.70 | 75.75 | 78.93 | 83.74 | 79.23 | 23.92 |
| **Logistic** | **18.64** $_{34.4\%↓}$ | **24.81** $_{9.3\%↑}$ | **56.79** $_{25.0\%↓}$ | **48.96** $_{38.0\%↓}$ | **85.84** $_{2.5\%↑}$ | **80.81** $_{1.2\%↑}$ | 24.77 $_{2.4\%↓}$ |
| **Non-ridig Editing (with *Masactrl*)** | | | | | | | |
| Linear | 30.83 | 21.15 | 113.03 | 107.46 | 77.23 | 83.13 | **22.65** |
| Cosine | 22.40 | 22.91 | 90.75 | 82.05 | 79.32 | 83.33 | 21.81 |
| **Logistic** | **15.87** $_{29.2\%↓}$ | **24.66** $_{7.7\%↑}$ | **75.18** $_{17.2\%↓}$ | **59.22** $_{27.8\%↓}$ | **81.11** $_{2.3\%↑}$ | **84.32** $_{0.1\%↑}$ | 22.30 $_{1.5\%↓}$ |

## 5.2 Qualitative and Quantitative Comparison

**Qualitative Comparison.** As shown in Fig. 5, our *Logistic Schedule* demonstrates superior content preservation in each task. In tasks requiring fine-grained editing, such as attributes editing, the *Logistic Schedule* better preserves other attributes while making the desired changes. For tasks involving high-level semantics, such as object translation and style/scene transfer, the *Logistic Schedule* maintains the overall structure and pose more effectively. In tasks that involve low-level semantics like color and texture, such as pose and attributes editing, the *Logistic Schedule* shows better fidelity and consistency. For tasks that require background preservation, such as object translation and pose editing, the *Logistic Schedule* excels in maintaining the background integrity. Overall, the *Logistic Schedule* ensures higher edit fidelity across various tasks, whereas the linear and cosine schedules sometimes fail to maintain the desired quality and consistency.

**Quantitative Comparison.** Table 1 shows that when employing the *Logistic Schedule*, all editing tasks exhibit improved retention of background and overall structure. While in some situations, the *Logistic Schedule* achieves slightly lower text alignment than the linear schedule, its preservation of background and structure is significantly superior.

## 5.3 Ablation Studies

In this section, we investigate the effects of different configurations of the Logistic Schedule and the adaptability of the Logistic Schedule with various inversion techniques and diffusion models. More experiments on hyperparameters (*e.g.*, guidance scale, input scale) can be found in *Appendix* E. The comparison with more design of the noise scheduler is provided in *Appendix* E.4.

### 5.3.1 Effects of Configuration of Logistic Schedule

We conduct experiments with different configurations of the Logistic Schedule in Eq. 5, providing further evidence for the noise space analysis (Section 4.2). The parameters of the Logistic Schedule (Eq. 5)—specifically the steepness ($k$) and the midpoint ($t_0$)—play a crucial role in balancing content preservation and edit fidelity. Table 2 provides the quantitative results of varying $k$ and $t_0$.

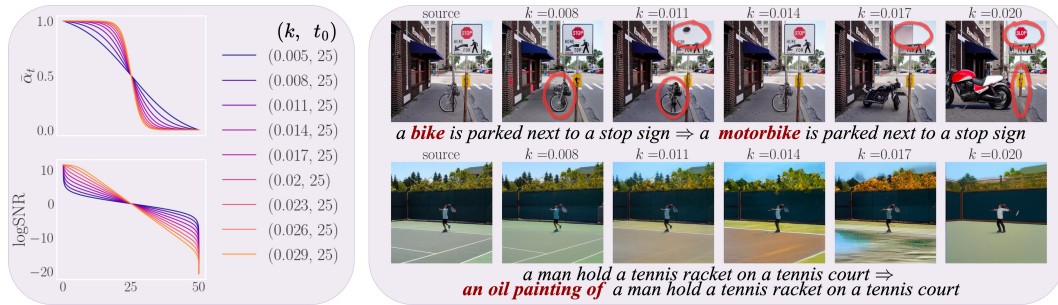

Figure 6: **Impact of $k$ on the logistic schedule**. Left: change in $\bar{\alpha}_t$ and logSNR with different $k$ values. Right: the effect of $k$ on edited images.

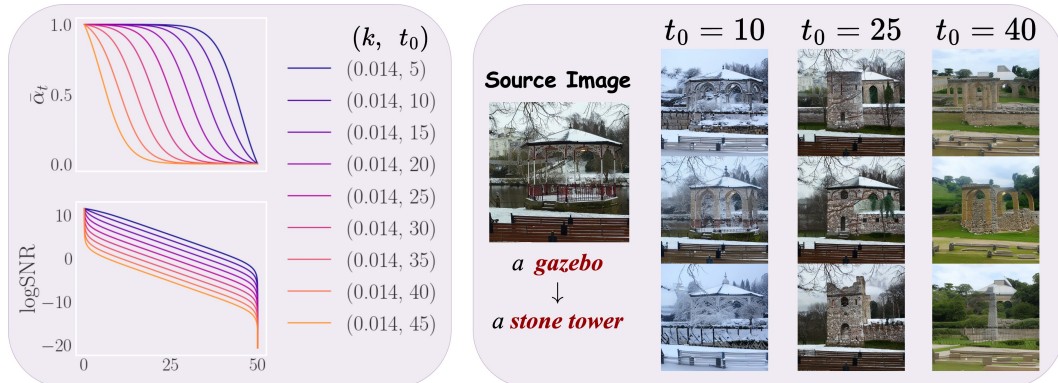

Figure 7: **Impact of $t_0$ on the logistic schedule**. Left: change in $\bar{\alpha}_t$ and logSNR with different $t_0$ values. Right: each column represents edited results within three random seeds, under a specific $t_0$.

Table 2: Quantitative results of the Logistic Schedule across various hyperparameter settings. The best method is indicated in bold, and the worst method is shown in purple.

| Settings | Structure Dist $_{\times 10^{-3}}$ ↓ | Background Preservation | | | | CLIP Similarity (%) | |
|---|---|---|---|---|---|---|---|
| | | PSNR ↑ | LPIPS $_{\times 10^{-3}}$ ↓ | MSE $_{\times 10^{-4}}$ ↓ | SSIM $_{\times 10^{-2}}$ ↑ | Visual ↑ | Textual ↑ |
| $k = 0.015$ $t_0 = \mathrm{int}(0.6T)$ | **17.37** | 24.78 | 81.80 | 49.47 | 82.97 | 82.44 | 23.62 |
| $k = 0.008$ | 16.27 | **26.45** | **75.62** | **43.20** | **85.28** | **84.07** | 20.47 |
| $k = 0.011$ | 16.64 | 25.80 | 77.90 | 48.83 | 84.15 | 83.76 | 21.46 |
| $k = 0.017$ | 22.79 | 22.33 | 99.98 | 57.32 | 81.52 | 82.10 | 23.25 |
| $k = 0.029$ | 27.82 | 21.05 | 103.45 | 64.36 | 78.48 | 80.66 | 23.81 |
| $t_0 = \mathrm{int}(0.4T)$ | 24.31 | 22.41 | 97.21 | 60.84 | 79.72 | 79.47 | 20.33 |
| $t_0 = \mathrm{int}(0.8T)$ | 29.47 | 21.64 | 95.58 | 63.89 | 75.14 | 77.05 | 22.68 |

**Different $k$: Changing the steepness of logSNR.** When $k$ is larger, the logSNR values span a larger range (Fig.6, left). However, if the range is too large, excessive steepness of logSNR results in excessive loss of original image information in edited images (Fig.6, right). Interestingly, when $k$ is small, the logSNR resembles that of linear and cosine schedules, but the logistic schedule better preserves the original image content without altering the overall structure. This further supports Proposition 3.1 that the singularity in linear and cosine schedules tends to destroy original image information, causing undesired changes.

**Different $t_0$: Introducing shifts in logSNR.** When $t_0$ is close to 0, the lower bound of logSNR is higher, affecting editability by reducing diversity and fidelity, as shown in Fig. 7. Conversely, when $t_0$ is close to $T$, the original information is lost too quickly, degrading content preservation.

Balancing these parameters, we find that $k = 0.015$ and $t_0 = \mathrm{int}(0.6T)$ strike the optimal trade-off between content preservation and edit fidelity, providing robust performance across various tasks.

### 5.3.2 Adapting Inversion Techniques and Diffusion Models

To validate the adaptability and robustness of the *Logistic Schedule*, we first apply it with Plug-and-Play [59] using Stable Diffusion v1.5 [50] as the baseline. We then design experiments with two other diffusion models, Stable Diffusion v2.1 and Stable Diffusion XL [47], and incorporate three advanced inversion approaches: Null-Text Inversion [41], Negative Prompt Inversion (NPI) [40], and Direct Inversion [25]. As shown in Table 3, while more advanced stable diffusion models increase textual similarity, they degrade content preservation. Conversely, incorporating advanced inversion approaches improves both content preservation and edit fidelity. Furthermore, Table 4 presents the

Table 3: Comparative performance metrics with different base models and inversion techniques.

| Variants | Structure Dist $_{\times 10^{-3}}$ ↓ | PSNR ↑ | Background Preservation LPIPS $_{\times 10^{-3}}$ ↓ | MSE $_{\times 10^{-4}}$ ↓ | SSIM $_{\times 10^{-2}}$ ↑ | CLIP Similarity Visual ↑ | Textual ↑ |
|---|---|---|---|---|---|---|---|
| **PnP+SD-1.5** | 26.66 | 22.46 | 111.27 | 77.74 | 80.02 | 81.24 | 21.74 |
| **Changing the Base Model** | | | | | | | |
| **SD-2.1** | 34.74 $_{30.3\%↑}$ | 19.94 $_{11.2\%↓}$ | 152.25 $_{36.8\%↑}$ | 122.86 $_{58.0\%↑}$ | 74.41 $_{7.0\%↓}$ | 79.38 $_{1.5\%↓}$ | 22.87 $_{5.2\%↑}$ |
| **SDXL** | 28.33 $_{6.3\%↑}$ | 21.57 $_{4.0\%↓}$ | 122.14 $_{9.8\%↑}$ | 89.02 $_{14.5\%↑}$ | 77.25 $_{3.5\%↓}$ | 77.52 $_{2.9\%↓}$ | **23.59** $_{8.5\%↑}$ |
| **Incorporating Advanced Inversion Approaches** | | | | | | | |
| **+ Null-Text** | 18.67 $_{30.0\%↓}$ | 23.80 $_{6.0\%↑}$ | 89.64 $_{19.4\%↓}$ | 57.97 $_{25.4\%↓}$ | 82.97 $_{3.7\%↑}$ | 82.46 $_{1.0\%↑}$ | 21.95 $_{1.0\%↑}$ |
| **+ NPI** | 24.82 $_{6.9\%↓}$ | 23.17 $_{3.2\%↑}$ | 99.19 $_{10.9\%↓}$ | 71.24 $_{8.4\%↓}$ | 80.26 $_{0.3\%↑}$ | 80.16 $_{0.9\%↓}$ | 21.53 $_{1.0\%↓}$ |
| **+ Direct** | **16.06** $_{39.8\%↓}$ | **25.73** $_{14.6\%↑}$ | **74.17** $_{33.3\%↓}$ | **41.19** $_{47.0\%↓}$ | **85.61** $_{7.0\%↑}$ | **83.29** $_{1.6\%↑}$ | 22.03 $_{1.3\%↑}$ |

detailed comparison between the *Logistic Schedule* and the scaled linear schedule across different inversion techniques.

Table 4: Comparison of inversion techniques with the scaled linear schedule and our proposed Logistic Schedule.

| Variants | Structure Dist ↓ | PSNR ↑ | Background Preservation LPIPS ↓ | MSE ↓ | SSIM ↑ | CLIP Similarity Visual ↑ | Textual ↑ |
|---|---|---|---|---|---|---|---|
| Null-Text + Linear | 21.00 | 23.00 | 95.00 | 63.00 | 81.50 | 81.00 | 21.30 |
| Null-Text + Logistic | 18.67 | 23.80 | 89.64 | 57.97 | 82.97 | 82.46 | 21.95 |
| NPI + Linear | 28.00 | 22.40 | 105.00 | 78.00 | 78.50 | 78.50 | 20.90 |
| NPI + Logistic | 24.82 | 23.17 | 99.19 | 71.24 | 80.26 | 80.16 | 21.53 |
| Direct + Linear | 19.00 | 24.90 | 78.00 | 45.00 | 83.50 | 82.90 | 21.90 |
| Direct + Logistic | 16.06 | 25.73 | 74.17 | 41.19 | 85.61 | 83.29 | 22.03 |

## 6 Conclusion

This paper presents the *Logistic Schedule*, a novel noise schedule that eliminates singularities and improves inversion stability for image editing. Our method enhances content preservation and edit fidelity without requiring additional retraining, making it a plug-and-play solution for existing workflows. Through in-depth analysis of the diffusion inversion process, we identify that current schedulers suffer from singularity issues at the start of inversion. The proposed Logistic Schedule provides a straightforward solution to this problem, offering superior performance and adaptability across various image editing tasks.

## 7 Acknowledgement

This work was supported by National Natural Science Foundation of China (62403429, 62293551, 62377038, 62177038, 62277042). Project of China Knowledge Centre for Engineering Science and Technology, Project of Chinese academy of engineering "The Online and Offline Mixed Educational Service System for 'The Belt and Road' Training in MOOC China". "LENOVO-XJTU" Intelligent Industry Joint Laboratory Project.

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

# Appendix

## Table of Contents

# A  Neural ODEs of DDIM

To support the analysis in Section 3 on the failure of DDIM in inversion, we present the following connections to neural ODEs and DDIM.

## A.1  Preliminaries: Score-Based Generative Modeling with SDEs

We beginning with the process of constructing a diffusion process using SDEs, extending DDPM to infinite noise scales for evolving data distributions from initial to prior distributions.

**Perturbing Process with SDEs.** DDPM [18] sets noise scales so that $\mathbf{x}_T$ approximates $\mathcal{N}(0, \mathbf{I})$, leveraging multiple noise scales for success. Song et al. extended this to infinite noise scales, evolving the data distribution via an SDE. The goal is to construct a diffusion process $\{\mathbf{x}(t)\}_{t=0}^{T}$, where $\mathbf{x}(0) \sim p_0$ (data distribution) and $\mathbf{x}(T) \sim p_T$ (prior distribution). The process is modeled by the SDE:

$$\mathrm{d}\mathbf{x} = \mathbf{f}(\mathbf{x}, t)\mathrm{d}t + g(t)\mathrm{d}\mathbf{w} \tag{6}$$

where $\mathbf{w}$ is the standard Wiener process with time flowing backwards from $T$ to $0$, $\mathbf{f}(\cdot, t)$ is the drift coefficient, and $g(\cdot)$ is the diffusion coefficient.

**Generating Samples by Reversing the SDE.** Starting from $\mathbf{x}(T) \sim p_T$ and reversing the process, we can obtain $\mathbf{x}(0) \sim p_0$, given by the reverse-time SDE:

$$\mathrm{d}\mathbf{x} = \left[\mathbf{f}(\mathbf{x}, t) - g(t)^2 \nabla_{\mathbf{x}} \log p_t(\mathbf{x})\right] \mathrm{d}t + g(t)\mathrm{d}\mathbf{w}. \tag{7}$$

The score $\nabla_{\mathbf{x}} \log p_t(\mathbf{x})$ can be estimated by training a score-based model on samples using score matching [22, 56].

**Solving Reverse-Time SDE: Probability Flow ODE.** Numerical solvers approximate trajectories from SDEs. General-purpose methods like Euler-Maruyama and stochastic Runge-Kutta [31] discretize the stochastic dynamics. In addition to these, score-based models enable solving the reverse-time SDE via a deterministic process, known as the *probability flow ODE*:

$$\mathrm{d}\mathbf{x} = \left[\mathbf{f}(\mathbf{x}, t) - \frac{1}{2} g(t)^2 \nabla_{\mathbf{x}} \log p_t(\mathbf{x})\right] \mathrm{d}t \tag{8}$$

This ODE is determined from the SDE once scores are known. When the score function is approximated by a neural network, it exemplifies a neural ODE [8].

**From Score-Based Models to DDPM: VE, VP SDEs** The noise perturbations in SMLD [56] and DDPM [18] are discretizations of two SDEs: *Variance Exploding (VE)* SDE and *Variance Preserving (VP)* SDE.

For SMLD with $N$ noise scales, each perturbation kernel $p_{\sigma_i}(\mathbf{x} \mid \mathbf{x}_0)$ corresponds to the distribution of $\mathbf{x}_i$ in this Markov chain:

$$\mathbf{x}_i = \mathbf{x}_{i-1} + \sqrt{\sigma_i^2 - \sigma_{i-1}^2}\, \mathbf{z}_{i-1}, \quad i = 1, \cdots, N, \tag{9}$$

where $\mathbf{z}_{i-1} \sim \mathcal{N}(0, \mathbf{I})$ and $\sigma_0 = 0$. As $N \to \infty$, $\{\sigma_i\}_{i=1}^{N}$ becomes $\sigma(t)$, $\mathbf{z}_i$ becomes $\mathbf{z}(t)$, and the Markov chain $\{\mathbf{x}_i\}_{i=1}^{N}$ becomes a continuous stochastic process $\{\mathbf{x}(t)\}_{t=0}^{1}$, given by the SDE:

$$\mathrm{d}\mathbf{x} = \sqrt{\frac{\mathrm{d}[\sigma^2(t)]}{\mathrm{d}t}}\, \mathrm{d}\mathbf{w}. \tag{10}$$

For DDPM, the perturbation kernels $\{p_{\alpha_i}(\mathbf{x} \mid \mathbf{x}_0)\}_{i=1}^{N}$ follow this Markov chain:

$$\mathbf{x}_i = \sqrt{1 - \beta_i}\, \mathbf{x}_{i-1} + \sqrt{\beta_i}\, \mathbf{z}_{i-1}, \quad i = 1, \cdots, N. \tag{11}$$

As $N \to \infty$, this converges to the SDE:

$$\mathrm{d}\mathbf{x} = -\frac{1}{2} \beta(t)\mathbf{x}\mathrm{d}t + \sqrt{\beta(t)}\mathrm{d}\mathbf{w}. \tag{12}$$

Thus, noise perturbations in SMLD and DDPM correspond to the SDEs 10 and 12, respectively. Notably, the SDE 10 results in an exploding variance as $t \to \infty$, while the SDE 12 maintains a fixed variance of one, demonstrating the superiority of VP SDE for stable variance preservation.

## A.2 Rewrite the DDIM Process as ODEs

**DDIM's Local Linearization Assumption:** DDIM inversion for real images is unstable due to its reliance on a local linearization assumption at each step, leading to error accumulation and content loss. DDIM assumes that the denoising process in Eq. 2 is roughly invertible:

$$\mathbf{x}_t^* = \frac{\mathbf{x}_{t-\Delta t}^* - b_t \epsilon(\mathbf{x}_t^*, t)}{a_t} \approx \frac{\mathbf{x}_{t-\Delta t}^* - b_t \epsilon(\mathbf{x}_{t-\Delta t}^*, t)}{a_t},$$

where $a_t = \sqrt{\alpha_{t-\Delta t}/\alpha_t}$ and $b_t = -\sqrt{\alpha_{t-\Delta t}(1 - \alpha_t)/\alpha_t} + \sqrt{1 - \alpha_{t-\Delta t}}$. This assumes $\epsilon(\mathbf{x}_t^*, t) \approx \epsilon(\mathbf{x}_{t-\Delta t}^*, t)$, and inversion accuracy depends on this assumption. Moreover, estimating the "predicted $\mathbf{x}_0$" at the beginning ($t = 1$) lacks a simple expression for the posterior mean conditioned on $\mathbf{x}_t$. This deviating from the linearization assumption causes the interpolation to break down from the start, resulting in server error accumulation problem.

**Relevance to Neural ODEs:** Under this assumption, the DDIM iteration process (Eq. 2) can be rewritten in a format similar to Euler integration for solving ODEs:

$$\frac{\boldsymbol{x}_{t-\Delta t}}{\sqrt{\alpha_{t-\Delta t}}} = \frac{\boldsymbol{x}_t}{\sqrt{\alpha_t}} + \left( \sqrt{\frac{1 - \alpha_{t-\Delta t}}{\alpha_{t-\Delta t}}} - \sqrt{\frac{1 - \alpha_t}{\alpha_t}} \right) \epsilon_\theta^{(t)}(\boldsymbol{x}_t). \tag{13}$$

Reparameterizing $(\sqrt{1 - \alpha}/\sqrt{\alpha})$ with $\sigma$ and $(\mathbf{x}/\sqrt{\alpha})$ with $\bar{\mathbf{x}}$, in the continuous case, $\sigma$ and $\mathbf{x}$ are functions of $t$, with $\sigma : \mathbb{R}_{\geq 0} \to \mathbb{R}_{\geq 0}$ continuous and increasing, $\sigma(0) = 0$. Eq. 13 can be seen as an Euler method over the ODE:

$$d\bar{\mathbf{x}}(t) = \epsilon_\theta^{(t)} \left( \frac{\bar{\mathbf{x}}(t)}{\sqrt{\sigma^2 + 1}} \right) d\sigma(t), \tag{14}$$

which corresponds to the Eq. 10 of probability flow ODE. This suggests that with enough discretization steps and the optimal model $\epsilon_\theta^{(t)}$, the generation process Eq. 2 can be reversed, encoding $\mathbf{x}_0$ to $\mathbf{x}_T$ and simulating the reverse of the ODE in Eq. 14.

**Theorem A.1** (DDIM ODEs). *While the ODEs are equivalent, the sampling procedures differ significantly. The Euler method for the probability flow ODE updates:*

$$\frac{\mathbf{x}_{t-\Delta t}}{\sqrt{\alpha_{t-\Delta t}}} = \frac{\mathbf{x}_t}{\sqrt{\alpha_t}} + \frac{1}{2} \left( \frac{1 - \alpha_{t-\Delta t}}{\alpha_{t-\Delta t}} - \frac{1 - \alpha_t}{\alpha_t} \right) \cdot \sqrt{\frac{\alpha_t}{1 - \alpha_t}} \cdot \epsilon_\theta^{(t)}(\mathbf{x}_t), \tag{15}$$

*which is equivalent to Eq. 13 if $\alpha_t$ and $\alpha_{t-\Delta t}$ are close enough. However, achieving this closeness is challenging with fewer time steps, and an inferior model can exacerbate the errors from this assumption. Moreover, the Variance Exploding SDE (VE SDE) has inherent flaws compared to the Variance Preserving SDE (VP SDE). VE SDEs tend to increase variance exponentially, leading to instability and less accurate representations, whereas VP SDEs maintain a stable variance, ensuring a more consistent and reliable modeling process.*

Modeling with d$t$ in Euler steps, as done in the probability flow ODE, ensures that the step size directly correlates with the temporal evolution, maintaining the integrity of the stochastic process and providing a more faithful representation of the underlying data distribution over time. [55] state that the ODE of DDIM is a special case of the probability flow ODE (continuous-time analog of DDPM).

*Proof.* We consider $t$ as a continuous, independent "time" variable and $\mathbf{x}$ and $\alpha$ as functions of $t$. Let's reparameterize DDIM and VE-SDE using $\bar{\mathbf{x}}$ and $\sigma$:

$$\bar{\mathbf{x}}(t) = \bar{\mathbf{x}}(0) + \sigma(t)\epsilon, \quad \epsilon \sim \mathcal{N}(0, I),$$

for $t \in [0, \infty)$ and a continuous function $\sigma : \mathbb{R}_{\geq 0} \to \mathbb{R}_{\geq 0}$ where $\sigma(0) = 0$.

Define $\alpha(t)$ and $\mathbf{x}(t)$ for DDIM as:

$$\bar{\mathbf{x}}(t) = \frac{\mathbf{x}(t)}{\sqrt{\alpha(t)}}, \quad \sigma(t) = \sqrt{\frac{1 - \alpha(t)}{\alpha(t)}}.$$

This implies:

$$\mathbf{x}(t) = \frac{\bar{\mathbf{x}}(t)}{\sqrt{\sigma^2(t) + 1}}, \quad \alpha(t) = \frac{1}{1 + \sigma^2(t)}.$$

From Equation 1, noting $\alpha(0) = 1$:

$$\frac{\mathbf{x}(t)}{\sqrt{\alpha(t)}} = \frac{\mathbf{x}(0)}{\sqrt{\alpha(0)}} + \sqrt{\frac{1 - \alpha(t)}{\alpha(t)}}\epsilon,$$

which reparameterizes to:

$$\bar{\mathbf{x}}(t) = \bar{\mathbf{x}}(0) + \sigma(t)\epsilon.$$

**ODE form for DDIM:** Simplify Equation 13 to:

$$\bar{\mathbf{x}}(t - \Delta t) = \bar{\mathbf{x}}(t) + (\sigma(t - \Delta t) - \sigma(t)) \cdot \epsilon_\theta^{(t)}(x(t)).$$

Dividing by $-\Delta t$ and taking $\Delta t \to 0$:

$$\frac{d\bar{\mathbf{x}}(t)}{dt} = \frac{d\sigma(t)}{dt}\epsilon_\theta^{(t)}\left(\frac{\bar{\mathbf{x}}(t)}{\sqrt{\sigma^2(t) + 1}}\right), \tag{16}$$

matching Equation 14.

**ODE form for VE-SDE:** Define $p_t(\bar{\mathbf{x}})$ as the data distribution perturbed with $\sigma^2(t)$ Gaussian noise. The probability flow for VE-SDE is given by:

$$d\bar{\mathbf{x}} = -\frac{1}{2}g(t)^2\nabla_{\bar{\mathbf{x}}}\log p_t(\bar{\mathbf{x}})dt,$$

where $g(t) = \sqrt{\frac{d\sigma^2(t)}{dt}}$. The perturbed score function $\nabla_{\bar{\mathbf{x}}}\log p_t(\bar{\mathbf{x}})$ minimizes:

$$\nabla_{\bar{\mathbf{x}}}\log p_t = \arg\min_{g_t} \mathbb{E}_{x(0)\sim q(x),\epsilon\sim\mathcal{N}(0,I)}[\|g_t(\bar{\mathbf{x}}) + \epsilon/\sigma(t)\|_2^2],$$

where $\bar{\mathbf{x}} = \bar{\mathbf{x}}(t) + \sigma(t)\epsilon$.

The equivalence between $\mathbf{x}(t)$ and $\bar{\mathbf{x}}(t)$ gives:

$$\nabla_{\bar{\mathbf{x}}}\log p_t(\bar{\mathbf{x}}) = -\frac{\epsilon_\theta^{(t)}\left(\frac{\bar{\mathbf{x}}(t)}{\sqrt{\sigma^2(t)+1}}\right)}{\sigma(t)}.$$

Using Equation A.2, and the definition of $g(t)$:

$$\frac{d\bar{\mathbf{x}}(t)}{dt} = \frac{1}{2}\frac{d\sigma^2(t)}{dt}\frac{\epsilon_\theta^{(t)}\left(\frac{\bar{\mathbf{x}}(t)}{\sqrt{\sigma^2(t)+1}}\right)}{\sigma(t)}dt,$$

rearranging terms:

$$\frac{d\bar{\mathbf{x}}(t)}{dt} = \frac{d\sigma(t)}{dt}\epsilon_\theta^{(t)}\left(\frac{\bar{\mathbf{x}}(t)}{\sqrt{\sigma^2(t) + 1}}\right),$$

which matches Equation 16. Both initial conditions are $\bar{\mathbf{x}}(T) \sim \mathcal{N}(0, \sigma^2(T)I)$, showing that the ODEs are identical. □

However, the above proof is based on several assumptions as follows:

1. **Equivalence Between $\mathbf{x}(t)$ and $\bar{\mathbf{x}}(t)$:** The bijective mapping between the variables $\mathbf{x}(t)$ and $\bar{\mathbf{x}}(t)$ is crucial for transforming the DDIM formulation into the VE-SDE framework. If this equivalence does not hold perfectly, the transformation could introduce errors. Small discrepancies can accumulate over time, leading to significant deviations in the modeling process, resulting in unreliable outcomes.

2. **Gaussian and Constant Noise $\epsilon$:** The noise $\epsilon$ is assumed to be Gaussian $\mathcal{N}(0, I)$ and constant throughout the process, which simplifies the mathematical formulation and integration. However, in real-world scenarios, the noise might not be perfectly Gaussian or constant. Variations in the noise can affect the accuracy of the model's predictions, leading to inconsistencies and unreliable results.

3. **Continuity and Differentiability of $\alpha(t)$ and $\sigma(t)$:** The functions $\alpha(t)$ and $\sigma(t)$ are assumed to be continuous and differentiable. This ensures smooth transitions and allows for the derivation of the

differential equations. If $\alpha(t)$ or $\sigma(t)$ are not continuous or differentiable, the resulting differential equations may not accurately represent the underlying processes. This can lead to instability and errors in the model's behavior.

4. **Optimal Model $\epsilon_\theta^{(t)}$:** The model $\epsilon_\theta^{(t)}$ is assumed to be optimal, meaning it perfectly minimizes the given loss function. In practice, achieving an optimal model is challenging. Suboptimal models can lead to inaccuracies in the predictions, and the error can propagate, reducing the reliability of the entire process.

5. **Closeness of $\alpha_t$ and $\alpha_{t-\Delta t}$:** It is assumed that $\alpha_t$ and $\alpha_{t-\Delta t}$ are close enough, which is necessary for the equivalence between the DDIM and VE-SDE formulations to hold. With fewer time steps, this assumption may not hold, leading to significant errors. Additionally, if the model is inferior, the errors arising from this assumption can be magnified, resulting in an unreliable process.

## B   Proofs

In this section, we first provide the detailed expressions of $\mathbf{x}_t$ with respect to different noise schedules. Then we provide the proof of the singularities problem in Proposition 3.1.

By reparameterizing:

$$\alpha_t = 1 - \beta_t, \qquad \bar{\alpha}_t = \prod_{i=1}^{t} \alpha_i, \tag{17}$$

the forward process of DDPM can be expressed as:

$$\mathbf{x}_t = \sqrt{\bar{\alpha}_t}\mathbf{x}_0 + \sqrt{1 - \bar{\alpha}_t}\epsilon.$$

Apply the chain rule to $d\mathbf{x}_t/dt$, we get:

$$\frac{d\mathbf{x}_t}{dt} = \frac{1}{2}\frac{1}{\sqrt{\bar{\alpha}_t}}\mathbf{x}_0\frac{d\bar{\alpha}_t}{dt} + \frac{1}{2}\frac{-1}{\sqrt{1-\bar{\alpha}_t}}\epsilon\frac{d\bar{\alpha}_t}{dt} \tag{18}$$

### B.1   Proof Preliminaries

#### B.1.1   Scaled Linear Schedule

The linear beta schedule is defined by:

$$\beta_t = \beta_{\text{start}} + t \cdot \frac{\beta_{\text{end}} - \beta_{\text{start}}}{T - 1}$$

where

$$\beta_{\text{start}} = \frac{0.0001 \cdot 1000}{T} = \frac{0.1}{T}$$

and

$$\beta_{\text{end}} = \frac{0.02 \cdot 1000}{T} = \frac{20}{T}$$

Thus,

$$\beta_t = \frac{0.1}{T} + t \cdot \frac{\frac{20}{T} - \frac{0.1}{T}}{T - 1} = \frac{0.1}{T} + t \cdot \frac{19.9}{T(T - 1)}$$

In general form, the expression for $\beta_t$ is:

$$\beta_t = \frac{0.1}{T} + \frac{19.9 \cdot t}{T(T - 1)}, \quad t = 0, 1, 2, \ldots, T - 1$$

Incorporating Eq. 17, the $\bar{\alpha}_t$ of scaled linear schedule is given by:

$$\bar{\alpha}_t = \prod_{i=1}^{t} \left(1 - \frac{0.1}{T} - \frac{19.9 \cdot i}{T(T - 1)}\right) \tag{19}$$

### B.1.2 Cosine Schedule

The cosine schedule is proposed in the iDDPM [43], where the definition of the schedule is given by:

$$\bar{\alpha}_t = \frac{f(t)}{f(0)}, \quad f(t) = \cos\left(\frac{t/T + s}{1 + s} \cdot \frac{\pi}{2}\right)^2, \tag{20}$$

where $s$ is a small offset to prevent $\beta_t$ from being too small near $t = 0$. Nichol and Dhariwal chose this setting since they found that having tiny amounts of noise at the beginning of the process made it hard for the network to predict accurately enough. Specifically, $s$ is set as 0.008 such that $\sqrt{\beta_0}$ was slightly smaller than the pixel bin size 1/127.5.

Plugging $f(t)$ into the expression, we get:

$$\bar{\alpha}_t = \frac{\cos^2(\frac{t/T + s}{1 + s} \cdot \frac{\pi}{2})}{\cos^2(\frac{s}{1 + s} \cdot \frac{\pi}{2})}$$

### B.1.3 Sigmoid Schedule

The sigmoid schedule is introduced in Jabri et al., which is designed for scalable data generation, especially for high-dimensional data, without addressing the challenges of DDIM inversion. The formulation of the sigmoid schedule can be presented as below:

$$\tilde{\alpha}_t = \frac{-\left(\frac{t(e-s)+s}{r}\right) \cdot \text{sigmoid}() + v_e}{v_e - v_s} \tag{21}$$

where $s$ and $e$ are the start and end of the sigmoid function's range, and $v_s = (s/r) \cdot \text{sigmoid}()$, $v_e = (e/r) \cdot \text{sigmoid}()$.

### B.1.4 Logistic Schedule

Recall the expression of the logistic schedule in Eq. 5:

$$\bar{\alpha}_t = \text{Normalized}\left(\frac{1}{1 + e^{-k(t-t_0)}}\right),$$

where $k$ and $t$ are hyperparameters that control the steepness and midpoint of the logistic function, respectively.

### B.2 Derivation of Singularities *w.r.t.* Linear and Cosine Schedules

Recall the Proposition 3.1:

**Proposition B.1** (Singularity in Inversion Process)**.** *During the inversion process, there exists a singularity at t = 0 for both the scaled linear and cosine schedule:*

$$When\ t = 0,\ \frac{d\mathbf{x}_t}{dt}\bigg|_{t \to 0} = \frac{0}{0} \cdot sign(\epsilon) = \infty \cdot sign(\epsilon).$$

Next, we provide the derivatives for scaled linear and cosine in order, to support Proposition 3.1.

### B.2.1 Scaled Linear Schedule

For $d\mathbf{x}_t/dt$, where $\mathbf{x}_t = \sqrt{\bar{\alpha}_t}\mathbf{x}_0 + \sqrt{1 - \bar{\alpha}_t}\epsilon$, cannot use $\bar{\alpha}_t = \prod_{i=1}^{t}\left(1 - \frac{0.1}{T} - \frac{19.9 \cdot i}{T(T-1)}\right)$ to find the feasible derivatives. Since the expression $\bar{\alpha}_t = \prod_{i=1}^{t}\left(1 - \frac{0.1}{T} - \frac{19.9 \cdot i}{T(T-1)}\right)$ represents a product of terms, which makes it difficult to differentiate directly. Taking the derivative of a product involves applying the product rule multiple times, which becomes impractical as the number of terms increases. Instead, we can use logarithms to simplify the expression into a sum, which is easier to handle analytically. This approach allows us to find an analytic approximation for $\bar{\alpha}_t$ and subsequently for $d\mathbf{x}_t/dt$.

*Proof.* The logarithm of the product in Eq. 19 reads:

$$\log(\bar{\alpha}_t) = \sum_{i=1}^{t} \log\left(1 - \frac{0.1}{T} - \frac{19.9 \cdot i}{T(T-1)}\right)$$

Given the small terms $\frac{0.1}{T}$ and $\frac{19.9 \cdot i}{T(T-1)}$, we can consider using a first-order Taylor expansion for the logarithm around 1. The Taylor expansion of $\log(1-x)$ around $x = 0$ is $\log(1-x) \approx -x$ for small $x$. Substituting, we get:

$$\log(\bar{\alpha}_t) \approx -\sum_{i=1}^{t} \left(\frac{0.1}{T} + \frac{19.9 \cdot i}{T(T-1)}\right)$$

Plugging $\sum_{i=1}^{t} i = \frac{t(t+1)}{2}$ into the expression, we get:

$$\log(\bar{\alpha}_t) \approx -\sum_{i=1}^{t} \frac{0.1}{T} - \sum_{i=1}^{t} \frac{19.9 \cdot i}{T(T-1)} = -\frac{0.1t}{T} - \frac{19.9}{T(T-1)} \cdot \frac{t(t+1)}{2}$$

Let:

$$f(t) = -\frac{0.1t}{T} - \frac{19.9t(t+1)}{2T(T-1)}$$

We have:

$$f'(t) = -\frac{0.1}{T} - \frac{19.9}{2T(T-1)}(2t+1) = -\frac{0.1}{T} - \frac{19.9(2t+1)}{2T(T-1)}$$

Plug $\bar{\alpha}_t = e^{f(t)}$ into the chain rule of $\frac{d\bar{\alpha}_t}{dt}$ and substituting $f(t)$ and $f'(t)$, we have:

$$\frac{d\bar{\alpha}_t}{dt} = \frac{d}{dt}\left(e^{f(t)}\right) = e^{f(t)} \cdot f'(t)$$

$$= \exp\left(-\frac{0.1t}{T} - \frac{19.9t(t+1)}{2T(T-1)}\right) \cdot \left(-\frac{0.1}{T} - \frac{19.9(2t+1)}{2T(T-1)}\right)$$

Substituting $\frac{d\bar{\alpha}_t}{dt}$ back into the expression for $\frac{dx_t}{dt}$ in Eq. 18:

$$\frac{dx_t}{dt} = \frac{\left(\epsilon e^{-\frac{t(0.1T+9.95t+9.85)}{T(T-1)}} - x_0\sqrt{1 - e^{-\frac{t(0.1T+9.95t+0.85)}{T(T-1)}}}\sqrt{e^{-\frac{t(0.1T+9.95t+9.85)}{T(TT1)}}}\right)(0.1T + 19.9t + 9.85)}{2T\sqrt{1 - e^{-\frac{t(0.1T+9.95t+9.85)}{T(T-1)}}}(T-1)} \tag{22}$$

So, we have:

$$\text{When } t = 0, \quad \frac{dx_t}{dt}\bigg|_{t\to 0} = \frac{\tilde{\infty}\epsilon(0.1T + 9.85)}{T(T-1)} = \tilde{\infty} \cdot \text{sign}(\epsilon).$$

where $\tilde{\infty}$ denotes an unspecified directed infinity in the complex plane.

$\square$

### B.2.2 Cosine Schedule

*Proof.* Given the expression:

$$x_t = \sqrt{\bar{\alpha}_t}x_0 + \sqrt{1 - \bar{\alpha}_t}\epsilon,$$

where:

$$\bar{\alpha}_t = \frac{\cos^2\left(\frac{t/T+s}{1+s} \cdot \frac{\pi}{2}\right)}{\cos^2\left(\frac{s}{1+s} \cdot \frac{\pi}{2}\right)}$$

By differentiating $\bar{\alpha}_t$, we have:

$$\frac{\mathrm{d}\bar{\alpha}_t}{\mathrm{d}t} = \frac{\left(2\cos\left(\frac{t/T+s}{1+s}\cdot\frac{\pi}{2}\right)\left(-\sin\left(\frac{t/T+s}{1+s}\cdot\frac{\pi}{2}\right)\right)\cdot\frac{\pi}{2T(1+s)}\right)\cos^2\left(\frac{s}{1+s}\cdot\frac{\pi}{2}\right)}{\cos^4\left(\frac{s}{1+s}\cdot\frac{\pi}{2}\right)}$$

Simplifying the expression:

$$\frac{\mathrm{d}\bar{\alpha}_t}{\mathrm{d}t} = \frac{2\cos\left(\frac{t/T+s}{1+s}\cdot\frac{\pi}{2}\right)\left(-\sin\left(\frac{t/T+s}{1+s}\cdot\frac{\pi}{2}\right)\right)\cdot\frac{\pi}{2T(1+s)}}{\cos^2\left(\frac{s}{1+s}\cdot\frac{\pi}{2}\right)}$$

Substituting $\dfrac{\mathrm{d}\bar{\alpha}_t}{\mathrm{d}t}$ back into the expression for $\dfrac{\mathrm{d}\mathbf{x}_t}{\mathrm{d}t}$ in Eq. 18:

$$\frac{\mathrm{d}\mathbf{x}_t}{\mathrm{d}t} = \frac{1}{2}\left(\frac{1}{\sqrt{\bar{\alpha}_t}}\mathbf{x}_0 - \frac{1}{\sqrt{1-\bar{\alpha}_t}}\epsilon\right)\cdot 2\cos\left(\frac{t/T+s}{1+s}\cdot\frac{\pi}{2}\right)\left(-\sin\left(\frac{t/T+s}{1+s}\cdot\frac{\pi}{2}\right)\right)\cdot\frac{\pi}{2T(1+s)}\cdot\frac{1}{\cos^2\left(\frac{s}{1+s}\cdot\frac{\pi}{2}\right)} \tag{23}$$

Considering the special cases:

- When $t = 0$, we have:

$$\frac{\mathrm{d}\mathbf{x}_0}{\mathrm{d}t} = 1.0(\infty\cdot\epsilon - 0.5\pi\mathbf{x}_0)\cdot\frac{\tan\left(\pi\frac{s}{2(1+s)}\right)}{T(1+s)}$$

- When $t = T$, we have:

$$\frac{\mathrm{d}\mathbf{x}_T}{\mathrm{d}t} = 0$$

□

### B.2.3 Sigmoid Schedule

*Proof.* Given definition of $x_t$:

$$\mathbf{x}_t = \sqrt{\bar{\alpha}_t}\mathbf{x}_0 + \sqrt{1-\bar{\alpha}_t}\epsilon_t$$

To express the coefficient of $\epsilon$ in the derivative of $\mathbf{x}_t$ with respect to $t$, we start with the expression:

$$\mathbf{x}_t = a(t)\cdot\epsilon + b(t)\cdot\mathbf{x}_0,$$

where $\epsilon$ represents the noise, and $\mathbf{x}_0$ is the original image. Given that $\epsilon$ and $\mathbf{x}_0$ are constants with respect to $t$, the differentiation yields:

$$\frac{\mathrm{d}}{\mathrm{d}t}\mathbf{x}_t = \epsilon\cdot\frac{\mathrm{d}}{\mathrm{d}t}a(t) + \mathbf{x}_0\cdot\frac{\mathrm{d}}{\mathrm{d}t}b(t). \tag{24}$$

Recall the definition of $\bar{\alpha}_t$ in sigmoid schedule in Eq. 21, we put it in the expression of $a(t)$, then the coefficient of $\epsilon$ in Eq. 24 can be expressed as:

$$\frac{\mathrm{d}}{\mathrm{d}t}a(t) = \frac{(e-s)e^{-s-t(e-s)}}{2\tau}\left(\sqrt{1-\left(\frac{\frac{1}{1+e^{\frac{s+t(e-s)}{\tau}}}-\frac{1}{1+e^{\frac{t(e-s)}{\tau}}}}{\frac{1}{1+e^{\frac{s}{\tau}}}-\frac{1}{1+e^{\frac{t(e-s)}{\tau}}}}\right)^2}\left(e^{-s-t(e-s)}+1\right)^2\right)^{-1}$$

When $t \to 0$, we have the derivative diverges to infinity:

$$\lim_{t\to 0}\frac{\mathrm{d}}{\mathrm{d}t}a(t) \to \infty,$$

□

## B.3 Derivatives of the Logistic Schedule

*Proof.* The $\bar{\alpha}_t$ given by the *Logistic Schedule* is:

$$\bar{\alpha}_t = \frac{1}{1 + e^{-k(t-t_0)}}$$

By differentiating $\bar{\alpha}_t$, we have:

$$\frac{d\bar{\alpha}_t}{dt} = \frac{d}{dt}\left(\frac{1}{1 + e^{-k(t-t_0)}}\right) = \frac{ke^{-k(t-t_0)}}{\left(1 + e^{-k(t-t_0)}\right)^2}$$

Substituting $\dfrac{d\bar{\alpha}_t}{dt}$ back into the expression for $\dfrac{d\mathbf{x}_t}{dt}$ in Eq. 18:

$$\frac{d\mathbf{x}_t}{dt} = \frac{1}{2}\left(\frac{1}{\sqrt{\bar{\alpha}_t}}\mathbf{x}_0 - \frac{1}{\sqrt{1-\bar{\alpha}_t}}\epsilon\right)\cdot\frac{ke^{-k(t-t_0)}}{\left(1 + e^{-k(t-t_0)}\right)^2}$$

Substitute $\bar{\alpha}_t$ back into the expression:

$$\frac{d\mathbf{x}_t}{dt} = \frac{ke^{-k(t-t_0)}}{2\left(1 + e^{-k(t-t_0)}\right)^2}\left(\frac{\mathbf{x}_0}{\sqrt{\frac{1}{1+e^{-k(t-t_0)}}}} - \frac{\epsilon}{\sqrt{1 - \frac{1}{1+e^{-k(t-t_0)}}}}\right) \tag{25}$$

$$= \frac{ke^{-k(t-t_0)}}{2\left(1 + e^{-k(t-t_0)}\right)^2}\left(\mathbf{x}_0\sqrt{1 + e^{-k(t-t_0)}} - \epsilon\sqrt{\frac{e^{-k(t-t_0)}}{1 + e^{-k(t-t_0)}}}\right) \tag{26}$$

When $t \to 0$:

$$\left.\frac{d\mathbf{x}_t}{dt}\right|_{t\to 0} = \frac{0.5\epsilon k\left(\frac{1}{e^{kt_0}+1.0}\right)^{0.5}e^{kt_0}}{e^{kt_0} + 1.0} - \frac{0.5kx_0e^{kt_0}}{\left(1.0 - \frac{1}{e^{kt_0}+1.0}\right)^{0.5}\left(e^{kt_0} + 1.0\right)^2}$$

Substitute the setting $k = 0.015, t_0 = \text{int}(0.3T)$ and $T = 100$ into the expression, we have:

$$\left.\frac{d\mathbf{x}_t}{dt}\right|_{t\to 0} = 1.486e^{-3}\epsilon - 1.318e^{-3}\mathbf{x}_0$$

$\square$

## C  Related Works

**Text-guided Image Editing.** Text-guided image editing significantly enhances the controllability and accessibility of visual manipulation by following human commands. With the advancement of large-scale training, diffusion models [49, 52, 50] have shown remarkable capabilities in transforming images based on human-given instructions [13, 62, 70]. Some approaches train end-to-end models for image editing [5, 28], while others propose training-free methods that merge information from source and target images using masks for controllability [39, 1]. A breakthrough by Hertz et al. leveraged the attention maps within the UNet to eliminate the need for manual masks, achieving promising results. This insight has been adopted and improved upon across multiple tasks by several works [59, 5, 6, 15, 73]. However, most current image editing approaches still rely on predefined noise schedules without evaluating their effectiveness. In this paper, we propose a newly designed noise schedule for image editing that provides high content preservation and enhanced editability.

**Inversion-based Image Editing.** Editing real images requires first inverting the image back to the latent space of the diffusion model due to the lack of a native latent space for these real images [46, 53, 65, 74], a process called image inversion. To address this, DDIM [55] introduced a deterministic sampling process for diffusion, allowing the inversion of the sampling process to recover the latent

noise. However, the invertible properties of DDIM rely on its linearization assumption, which introduces deviations that drive the inverted latent away from its true distribution. As the Markov properties of the diffusion process come into play, these deviations gradually enlarge, resulting in suboptimal inverted latents that degrade reconstruction and editing quality. Recently, several inversion-based methods have been proposed to mitigate this issue [41, 60, 40, 25]. These methods attempt to correct errors on the reconstruction path to the desired DDIM trajectory, ensuring that the original content in the source image is highly preserved and can be injected into the editing process for better content preservation. However, these methods still rely on the accuracy of DDIM inversion. This brings us to the root of the issue: correcting the DDIM errors themselves.

**Noise Schedule Adjustments.** Previous work on noise scheduling focuses on training diffusion models from scratch to improve image quality or optimize the variational lower bound [29, 23, 26, 14, 20, 34]. Hoogeboom et al. propose noise schedule adjustments and other strategies to effectively train standard denoising diffusion models on high-resolution images without additional sampling modifiers. Lin et al. reveal that common diffusion noise schedules fail to enforce zero terminal SNR, causing discrepancies between training and inference. However, none focus on designing an off-the-shelf noise schedule for image editing—a downstream task that does not require training from scratch and leverages existing models for sampling. This highlights the need for a simple but effective noise schedule tailored for downstream tasks like image editing.

# D  Experimental Settings

## D.1  Introduction of Editing Types

| Task Name | Source Prompt | Target Prompt |
|---|---|---|
| **Attributes Content** | a close up of a cat with yellow eyes | a close up of a cat with yellow eyes ***with its mouth open*** |
| **Attributes Color** | a smiling woman with ***brown*** eyes | a smiling woman with ***blue*** eyes |
| **Attributes Material** | a tiger is sitting in the grass | a ***silver tiger sculpture*** is sitting in the grass |
| **Object Switch** | ***bread*** on a table with tomatoes and a napkin | ***meat*** on a table with tomatoes and a napkin |
| **Object Addition** | a close up of a dog | a close up of a dog ***with sunglasses*** |
| **Non-Rigid Editing** | a tiger ***walking*** across a field in the wild | a tiger ***standing still*** on a field in the wild |
| **Scene Transferring** | a bench chair in front of ***mountains*** | a bench chair in front of ***the sea*** |
| **Style Transferring** | a man riding a skateboard on a ramp | a ***watercolor painting*** of a man riding a skateboard on a ramp |

Table 5: Editing tasks with example source and target prompts. The change parts are noted in red.

We conducts eight editing tasks based on the real images to verify the effectiveness and versatility, along with the corresponding challenges for each task (Table 5):

1. **Attributes Content Editing (Fig. 13)**: Modifies specific attributes, like changing a cat's expression. The **challenge** is ensuring high fidelity and preserving the original content without artifacts.
2. **Attributes Color Editing (Fig. 14)**: Alters color attributes, like changing a bird's color or eye color. The **challenge** is maintaining natural and coherent lighting and shading.
3. **Attributes Material Editing (Fig. 15)**: Changes material properties, like transforming a tiger into a silver sculpture. The **challenge** is accurately rendering new materials while preserving shape and avoiding unrealistic artifacts.
4. **Object Switch (Fig. 16)**: Switches objects, like replacing bread with meat or transforming a fox into a horse. The **challenge** is maintaining scene composition and seamlessly integrating new objects.

5. **Object Addition (Fig. 17)**: Adds new objects, like sunglasses to a dog or more strawberries in a bowl. The **challenge** is naturally integrating new objects, ensuring consistent lighting, shadows, and perspective.

6. **Non-Rigid Editing (Fig. 18)**: Makes non-rigid modifications, like changing the pose of a tiger. The **challenge** is preserving anatomical correctness and natural appearance while making pose changes.

7. **Scene Transferring (Fig. 19)**: Transfers scene context, like changing the background from mountains to the sea. The **challenge** is blending new backgrounds seamlessly with the foreground, maintaining consistent lighting, shadows, and color tones.

8. **Style Transferring (Fig. 20)**: Transfers artistic style, like converting a photo into a water-color painting. The **challenge** is preserving essential details and content while accurately applying the new artistic style.

## D.2 Implementation Details

All primary experiments are conducted using Stable Diffusion v1.5[2], with an image size of 512x512x3 and a latent space of 64x64x4. For ablation studies (Section 5.3.2), SD v2.1[3] and SDXL[4] are employed. Experiments run on a single Nvidia A100 GPU with 100 timesteps. The inversion (forward) guidance scale is set to 3.5, and the generation (reverse) guidance scale is set to 7.5. For the logistic schedule, $k$ is set to 0.015, and $t_0$ is set to $\text{int}(0.3T)$, where $T$ is the number of timesteps. Default hyperparameter settings are used unless otherwise specified. For each incorporated method, default hyperparameters are as follows:

- *Edit Friendly DDPM* [21]: $T_{\text{skip}} = 36$, starting generation from timestep $T - T_{\text{skip}}$.
- *StyleDiffusion* [63]: Uses SD v1.5 with 1000 inference timesteps and $T_{\text{trans}} = 301$ for style transfer.
- *MasaCtrl* [6]: Starts mutual self-attention control at step $S = 4$ and layer $L = 10$.
- *Pix2Pix Zero* [45]: Applies noise regularization for 5 iterations at each timestep with a weight $\lambda$ of 20.
- *Null-Text Inversion* [41]: 500 iterations for null-text optimization, with early stopping at $\epsilon = 1e^{-5}$.
- *Negative Prompt Inversion* [40]: Uses early stopping with a threshold increasing linearly from $1e^{-5}$ by a factor of $2e^{-5}$ through the sampling steps.

These settings ensure a consistent evaluation framework across different experiments and methods.

## D.3 Evaluation Metrics

In this work, seven metrics are employed to evaluate the effectiveness of the *Logistic Schedule*, including three aspects introduced below.

**Structure Distance:** Structure Distance: To evaluate the structure distance between the source and edited images, we leverage *DINO-I* [5], which was proposed by DreamBooth [51] to emphasize unique properties of identities. For *DINO-I*, we calculate the cosine similarity between the ViT-B/16 DINO [7] embeddings of source and generated images. Additionally, we consider fine-tuning and editing time as metrics to evaluate the efficiency of the editing process. Since DINO is trained in a self-supervised manner, it highlights general features rather than category-based distinctions, making it suitable for capturing the structural integrity of images.

**Background Preservation:** To measure how the background is preserved during editing, we apply *PSNR*, *LPIPS* [24, 72], *MSE*, and *SSIM* [64] in the area outside of the annotated masks. These metrics serve different roles in evaluating image quality and preservation:

---

[2]https://huggingface.co/runwayml/stable-diffusion-v1-5
[3]https://huggingface.co/stabilityai/stable-diffusion-2-1
[4]https://huggingface.co/stabilityai/stable-diffusion-xl-base-1.0
[5]https://huggingface.co/facebook/dino-vitb8

- *PSNR (Peak Signal-to-Noise Ratio)* measures the ratio between the maximum possible power of a signal and the power of corrupting noise, reflecting the overall quality of the image:

$$\text{PSNR} = 10 \cdot \log_{10}\left(\frac{\text{MAX}^2}{\text{MSE}}\right),$$

  where MAX is the maximum possible pixel value of the image (e.g., 255 for an 8-bit image), and MSE is the Mean Squared Error.
- *LPIPS (Learned Perceptual Image Patch Similarity)* evaluates perceptual similarity by comparing the differences in deep feature space using a pretrained deep network (such as VGG [54]), capturing human visual perception better than pixel-wise metrics, by calculating the Euclidean distance between the feature representations of two images:

$$\text{LPIPS} = \sum_l w_l \, \|\phi_l(x) - \phi_l(y)\|_2^2,$$

  where $\phi_l$ denotes the feature map at layer $l$ of the pretrained network, and $w_l$ are the weights for each layer.
- *MSE (Mean Squared Error)* calculates the average squared difference between original and edited image pixels, indicating the overall fidelity and error magnitude:

$$\text{MSE} = \frac{1}{N}\sum_{i=1}^{N}(I_1[i] - I_2[i])^2,$$

  where $N$ is the number of pixels in the image, $I_1$ and $I_2$ are the original and edited images respectively, and $i$ indexes the pixels.
- *SSIM (Structural Similarity Index Measure)* assesses image similarity by comparing luminance, contrast, and structure, providing a holistic view of image quality. The SSIM index between two images $x$ and $y$ is calculated as:

$$\text{SSIM}(x, y) = \frac{(2\mu_x\mu_y + C_1)(2\sigma_{xy} + C_2)}{(\mu_x^2 + \mu_y^2 + C_1)(\sigma_x^2 + \sigma_y^2 + C_2)},$$

  where $\mu_x$ and $\mu_y$ are the average pixel values of $x$ and $y$, $\sigma_x^2$ and $\sigma_y^2$ are the variances of $x$ and $y$, $\sigma_{xy}$ is the covariance of $x$ and $y$, and $C_1$ and $C_2$ are constants to stabilize the division when the denominator is close to zero.

Incorporating these metrics together can demonstrate background preservation more robustly since multiple metrics offer a comprehensive evaluation from different perspectives.

**Text-Image Consistency:** CLIP Similarity [48] evaluates the text-image consistency between the edited images and the corresponding target editing text prompts. *CLIP-I* and *CLIP-T* assess visual similarity and text-image alignment, respectively. For *CLIP-I*, we calculate the CLIP visual similarity between the source and generated images. For *CLIP-T*, we calculate the CLIP text-image similarity between the generated images and the given text prompts. These metrics help ensure that the edited images accurately reflect the intended modifications described in the text prompts.

## E  Experimental Results

### E.1  Quantitative Comparison Across Editing Types

We provide the performance of *Logistic Schedule* following the editing methods configuration in Section 5 in Table 6. The results vary across different editing types. Attributes content editing shows high PSNR and CLIP visual similarity. We attribute this to the relatively straightforward nature of modifying content attributes, which allows for high fidelity and coherence in the edited images compared to more complex edits. In the more challenging editing types, such as object addition (5th row) and non-rigid editing (e.g., pose, motion, 6th row), the model shows minimal changes, resulting in relatively better evaluation results in essential content preservation metrics. The limited alterations required in these tasks help maintain the original structure and details, leading to higher PSNR and SSIM values. Object switch shows high CLIP visual similarity. This can be attributed to the clear and distinct nature of object switching, which allows for more precise visual matching with the target

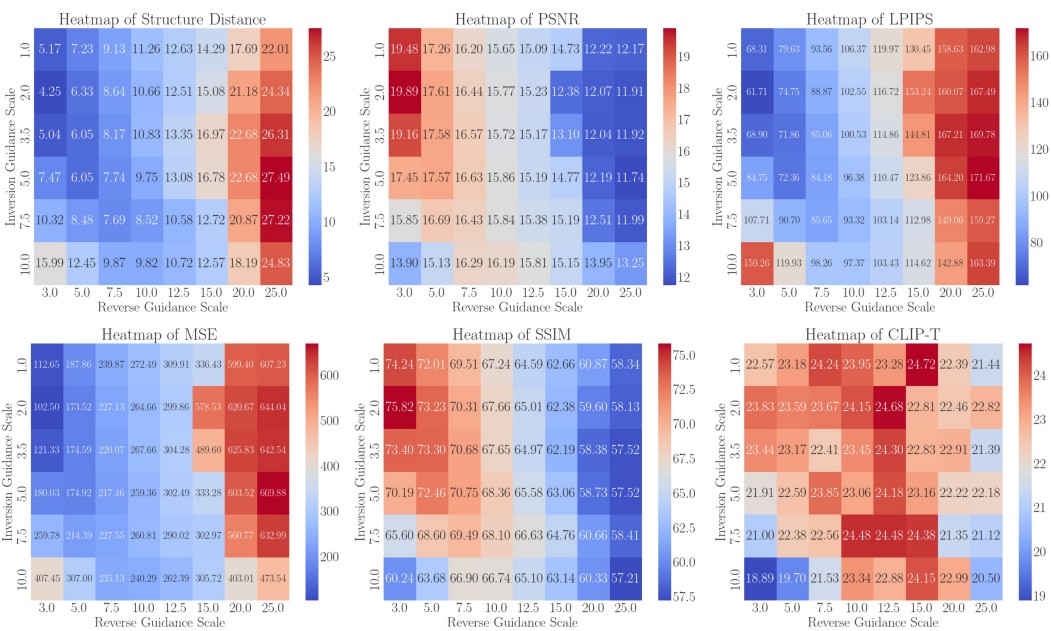

Figure 8: **Impact of different combinations of inversion and reverse guidance scales on various performance metrics**. Results are averaged on Attributes Editing tasks using Prompt-to-Prompt as the editing method [16], highlighting optimal scale settings for balanced performance across tasks.

Table 6: **Performance of *Logistic Schedule* on different editing tasks** in have ten independent runs with random seeds. Bold values indicate the best results, while underlined values denote the second-best results. 'Attr.', 'Obj.' and 'Trans.' denote 'Attributes', 'Objects', and 'Transferring', respectively.

| Edit Task | Structure Dist $_{\times 10^{-3}}$ ↓ | Background Preservation | | | | CLIP Similarity (%) | |
| --- | --- | --- | --- | --- | --- | --- | --- |
| | | PSNR ↑ | LPIPS $_{\times 10^{-3}}$ ↓ | MSE $_{\times 10^{-4}}$ ↓ | SSIM $_{\times 10^{-2}}$ ↑ | Visual ↑ | Textual ↑ |
| **Attr. Content** | $\underline{15.74}_{\pm 0.9}$ | $\mathbf{26.58}_{\pm 1.3}$ | $70.69_{\pm 2.1}$ | $51.04_{\pm 1.7}$ | $84.48_{\pm 3.0}$ | $\mathbf{89.30}_{\pm 3.2}$ | $22.05_{\pm 1.6}$ |
| **Attr. Color** | $16.78_{\pm 1.4}$ | $23.81_{\pm 1.2}$ | $89.65_{\pm 3.5}$ | $53.10_{\pm 1.1}$ | $81.04_{\pm 4.7}$ | $81.26_{\pm 2.3}$ | $19.41_{\pm 1.0}$ |
| **Attr. Material** | $18.67_{\pm 0.8}$ | $\underline{25.73}_{\pm 0.9}$ | $74.17_{\pm 3.4}$ | $\underline{41.19}_{\pm 1.6}$ | $81.61_{\pm 5.7}$ | $78.06_{\pm 3.3}$ | $\underline{24.03}_{\pm 1.5}$ |
| **Obj. Switch** | $22.40_{\pm 1.8}$ | $22.91_{\pm 1.2}$ | $90.75_{\pm 5.0}$ | $82.05_{\pm 1.4}$ | $79.32_{\pm 4.6}$ | $\underline{86.72}_{\pm 3.3}$ | $22.65_{\pm 1.9}$ |
| **Obj. Add** | $\mathbf{11.11}_{\pm 1.1}$ | $25.40_{\pm 1.5}$ | $63.05_{\pm 3.2}$ | $\mathbf{40.52}_{\pm 1.0}$ | $85.32_{\pm 4.6}$ | $76.33_{\pm 3.3}$ | $23.09_{\pm 0.9}$ |
| **Non-rigid** | $15.87_{\pm 1.7}$ | $24.66_{\pm 1.3}$ | $75.18_{\pm 2.5}$ | $59.22_{\pm 1.2}$ | $81.11_{\pm 4.8}$ | $81.26_{\pm 3.6}$ | $22.30_{\pm 1.1}$ |
| **Scene Trans.** | $17.63_{\pm 1.4}$ | $24.79_{\pm 1.3}$ | $\mathbf{55.57}_{\pm 2.3}$ | $48.51_{\pm 1.7}$ | $\mathbf{85.95}_{\pm 6.2}$ | $81.63_{\pm 1.6}$ | $22.11_{\pm 1.0}$ |
| **Style Trans.** | $19.66_{\pm 1.5}$ | $25.50_{\pm 1.8}$ | $\underline{60.24}_{\pm 4.2}$ | $49.79_{\pm 1.3}$ | $\underline{85.60}_{\pm 6.0}$ | $80.24_{\pm 5.6}$ | $\mathbf{25.73}_{\pm 1.1}$ |

object compared to other editing tasks. Style transferring (8th row) shows the highest CLIP text similarity. This is likely because the task involves applying well-defined artistic styles that closely align with the textual descriptions, resulting in edits that match the intended style effectively. Scene and style transferring (7th and 8th row) show high LPIPS and SSIM. We attribute this to the nature of the task, which involves changing backgrounds or styles while keeping the main subjects intact. This process maintains the overall scene coherence and structure consistency, resulting in enhanced perceptual quality and structural similarity.

## E.2 Qualitative Comparison Across Editing Types

The logistic noise schedule consistently outperforms linear and cosine schedules across various editing tasks. It excels in preserving the original image content while making specified changes without artifacts (Fig. 13), maintaining natural and coherent lighting in color edits (Fig. 14), and accurately rendering new material properties (Fig. 15). For object switching and addition, it ensures seamless integration with consistent lighting and spatial relationships (Figs. 16, 17). In non-rigid editing, it preserves anatomical correctness and smooth transitions (Fig. 18). It also blends new

backgrounds naturally in scene transfers (Fig. 19) and maintains essential details while applying new artistic styles (Fig. 20).

## E.3 Broader Application: Training-Based Methods

While the Logistic Schedule has shown broad applicability in various image editing tasks, we further explore its use in training-based methods, specifically in Text-to-Image synthesis. For this, we conducted experiments by fine-tuning the UNet using DreamBooth [51], leveraging approximately 100 images for each configuration.

Table 7: Performance comparison of DreamBooth fine-tuning using different noise schedules on the SD-1.5 model. The best results are in **bold**.

| Setting | DINO (↑) | CLIP-I (↑) | CLIP-T (↑) | PSNR (↑) | PRES (↑) |
|---|---|---|---|---|---|
| Real Images | 0.823 | 0.902 | N/A | 26.86 | 0.653 |
| DreamBooth (SD-1.5) + Linear | 0.726 | 0.823 | 0.268 | 24.41 | 0.567 |
| DreamBooth (SD-1.5) + Cosine | 0.745 | 0.857 | 0.264 | 24.75 | 0.554 |
| DreamBooth (SD-1.5) + **Logistic** | **0.761** | **0.877** | **0.293** | **25.62** | **0.580** |

The related qualitative comparisons are shown in Fig. 9. These results demonstrate that training DreamBooth with the Logistic Schedule improves performance across key metrics such as DINO and CLIP-I similarity, as well as PSNR and PRES, outperforming both linear and cosine schedules.

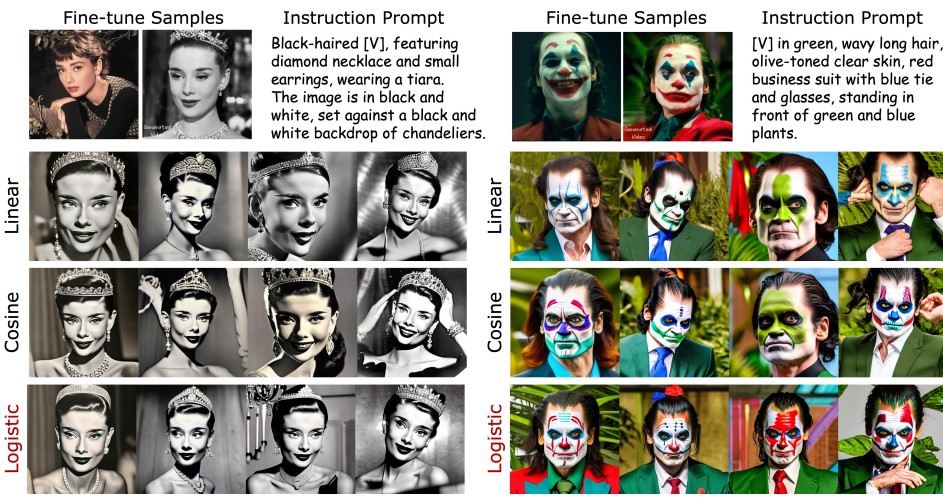

Figure 9: Qualitative comparisons of fine-tuning DreamBooth using different noise schedules (Linear, Cosine, and Logistic). The top column presents the fine-tune samples, and the instruction prompts, and the below column displays the corresponding fine-tuned outputs. The Logistic Schedule produces superior outputs with improved fidelity and alignment to the prompts.

## E.4 Comparison With Other Noise Schedulers

We conducted experiments comparing our Logistic Schedule with other schedules under the DDIM paradigm, such as exponential, sigmoid, hyperbolic, and geometric schedules. Table 8 displays the quantitative results. The best-performing method is indicated in **bold**, the worst method is marked in purple, and the second-best method is underlined.

The results demonstrate that our Logistic Schedule achieves competitive performance across various metrics. Notably, it offers significant improvements in content preservation and edit fidelity compared to other schedules. As shown in Fig. 10, the Logistic Schedule preserves the visual characteristics of the source image more faithfully during reconstruction and enables more precise control during editing.

Table 8: Comparison of different noise schedules. Metrics include Structure Distance ($\times 10^{-3}$), PSNR (higher is better), LPIPS ($\times 10^{-3}$, lower is better), MSE ($\times 10^{-4}$, lower is better), SSIM ($\times 10^{-2}$, higher is better), and CLIP Similarity for both visual and textual content.

| Schedule | Dist ↓ | PSNR ↑ | LPIPS ↓ | MSE ↓ | SSIM ↑ | Visual ↑ | Textual ↑ |
|---|---|---|---|---|---|---|---|
| Linear | 35.66 | 20.70 | 134.88 | 113.61 | 77.60 | 79.82 | 23.06 |
| Cosine | 26.57 | 22.38 | 110.52 | 80.01 | 80.15 | 81.35 | 22.39 |
| Exponential | 16.22 | 25.20 | 80.45 | 47.11 | 82.23 | 82.78 | 19.50 |
| Hyperbolic | 36.55 | 20.95 | 140.55 | 119.78 | 79.89 | 79.20 | 23.20 |
| Geometric | 18.12 | 24.10 | 92.45 | 62.13 | 82.05 | 82.20 | 20.45 |
| Sigmoid | 27.80 | 22.55 | 115.32 | 85.60 | 80.22 | 81.50 | 22.55 |
| Logistic | 17.37 | 24.78 | 81.80 | 49.47 | 82.97 | 82.44 | 23.62 |

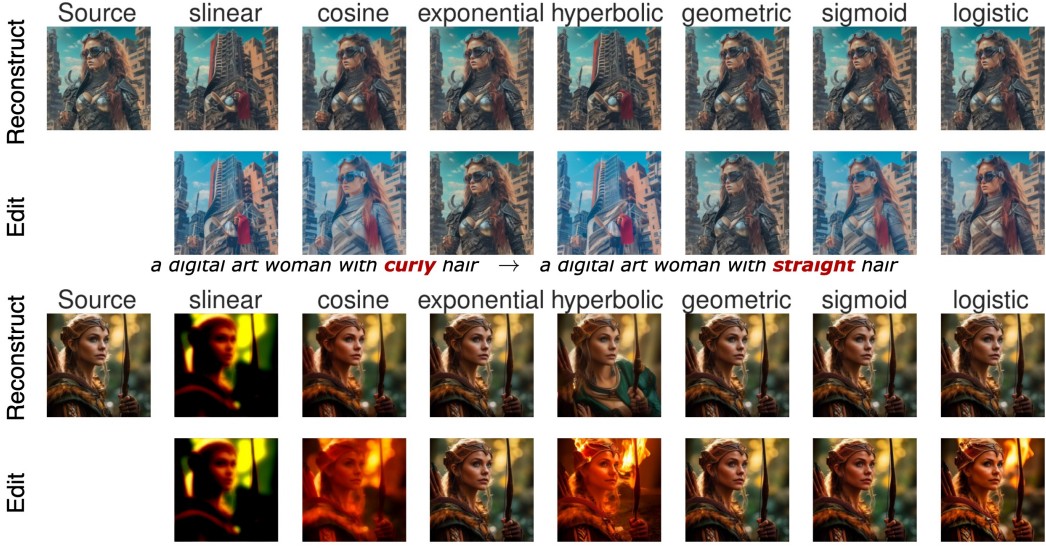

Figure 10: Qualitative comparison between different noise schedules for both reconstruction and editing tasks.

### E.5 Reconstruction Ability of Different Noise Schedule

Table 9: **Comparison of reconstruction quality** using different noise schedules for DDIM inversion and Direct Inversion, showing the superior performance of the Logistic Schedule.

| Inversion | Schedule | Structure | Background Preservation | | | |
|---|---|---|---|---|---|---|
| | | Dist $_{\times 10^{-3}}$ ↓ | PSNR ↑ | LPIPS $_{\times 10^{-3}}$ ↓ | MSE $_{\times 10^{-4}}$ ↓ | SSIM $_{\times 10^{-2}}$ ↑ |
| **DDIM Inversion** | **Linear** | 7.96 | 27.46 | 58.49 | 30.08 | 84.54 |
| | **Cosine** | 7.43 | 27.36 | 61.22 | 28.49 | 82.66 |
| | **Logistic** | 7.04 | 25.78 | 71.87 | 37.59 | 80.07 |
| **Direct Inversion** | **Linear** | 2.78 | 29.58 | 36.36 | 20.23 | 85.28 |
| | **Cosine** | 2.75 | 30.00 | 33.80 | 21.70 | 85.90 |
| | **Logistic** | 2.54 | 31.30 | 31.16 | 12.27 | 88.94 |

To demonstrate the ability of the *Logistic Schedule* to better align inversion by eliminating the singularity at the start point, we evaluate the reconstruction results of DDIM Inversion [55] and Direct Inversion [25] using scaled linear, cosine, and our logistic schedule. During reconstruction, the condition is the source prompt, which is also applied as a condition during inversion. The comparison of reconstruction quality is shown in Table 9.

Source Image

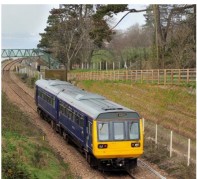

Source Prompt:
*a train traveling down the tracks*

Target Prompt:
*a **skyblue** train traveling down the tracks*

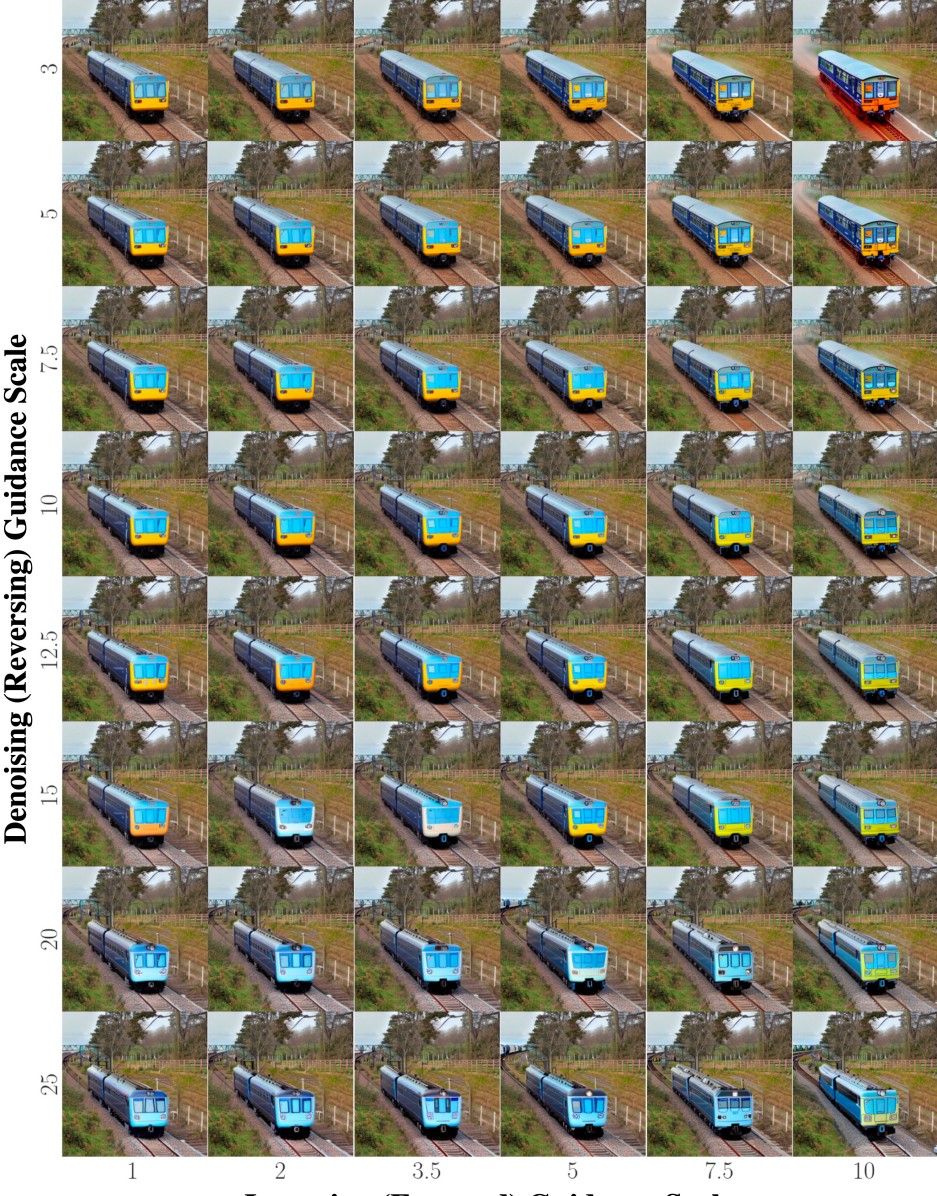

Figure 11: **Qualitative comparison of varying guidance scales during the inversion (forward) and denoising (reversing) processes of DDIM**. The guidance scales for inversion are varied across the columns (1 to 10), and the guidance scales for denoising are varied across the rows (3 to 25).

## E.6 Effects of Guidance Scale

We investigate the impact of the guidance scale on the inversion (forward) and generation (reserve) processes of DDIM with the *Logistic Schedule*, consequently affecting the editing results. We illustrate the impact of varying guidance scales during the inversion and denoising processes of DDIM on performance metrics in Fig.8, with an example of how the edited images are affected shown in Fig.11. When keeping the inverse guidance scale constant, we observed that as the reverse guidance scales increased gradually, background preservation initially decreased. The inflection point occurred when the inverse guidance scale equaled the forward guidance scale. In contrast, CLIP similarity showed a consistently increasing trend until the reverse guidance scale exceeded 15. The quantitative heatmaps and qualitative results highlight a noticeable trade-off between essential content preservation and edit fidelity. Optimal incorporation of both scales ensures a balance between structural preservation, perceptual quality, and text-image consistency, with a combination of 5.0 for inversion and 7.5 for reverse generally providing the best performance across most metrics. This trade-off arises because current editing methods struggle to differentiate between regions needing modification and those that do not, leading to substantial alterations of the source image and conflicting with content preservation objectives.

## E.7 Effects of Input Scale

Chen et al. proposed to modify noise scheduling by scaling the input $\mathbf{x}_0$ by a constant factor $b$ via:

$$\mathbf{x}_t = \sqrt{\bar{\alpha}_t} b \mathbf{x}_0 + \sqrt{1 - \bar{\alpha}_t} \epsilon.$$

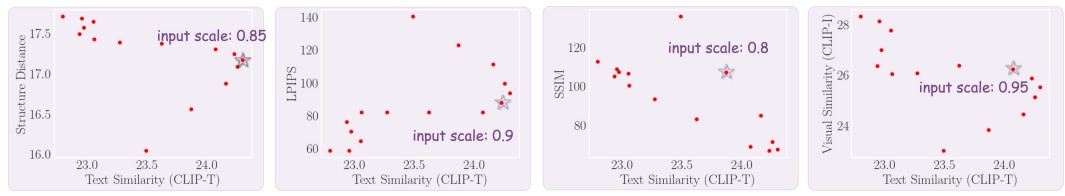

Figure 12: **Impact of input scale on content preservation and edit fidelity**. The optimal input scale balances the preservation of the original image structure (low structure distance, high SSIM) and the quality of the edits (high CLIP-T and CLIP-I).

As the scaling factor $b$ decreases, the original image's strength lessens and noise levels grow [9]. As previous image editing works have not extensively investigated the effects of input scale, we investigate the effects of input scale in image editing in this work. We change the input scale from 0.5 to 1.4 with a step size of 0.05, and illustrate the effects of the input scale on both content preservation and edit fidelity in Fig. 12. As observed, the input scale significantly impacts the balance between content preservation and edit fidelity. Higher input scales (closer to 1.4) better preserve the original image structure, as shown by lower structure distances and higher SSIM values but reduce edit fidelity (lower CLIP-T and CLIP-I scores). Conversely, lower input scales (closer to 0.5) enhance edit fidelity but degrade content preservation. The optimal input scale, found to be 0.8-0.95, achieves a balance between these objectives. This is because slightly higher noise levels improve editability while maintaining acceptable content preservation, providing a satisfactory trade-off in image editing.

Table 10: **Performance comparison of input scale normalization** in object switch task using Zero-shot Pix2Pix, showing no improvement in content preservation or edit fidelity.

| Input Scale | Structure Dist $_{\times 10^{-3}}$ ↓ | Background Preservation | | | | CLIP Similarity (%) | |
|---|---|---|---|---|---|---|---|
| | | PSNR ↑ | LPIPS $_{\times 10^{-3}}$ ↓ | MSE $_{\times 10^{-4}}$ ↓ | SSIM $_{\times 10^{-2}}$ ↑ | Visual ↑ | Textual ↑ |
| **w/o Normalizing** $b$ | 22.40 | 22.91 | 90.75 | 82.05 | 79.32 | 86.72 | 22.65 |
| **w. Normalizing** $b$ | 24.46 | 22.06 | 108.43 | 83.47 | 79.16 | 86.24 | 22.48 |

A strategy to improve training a diffusion model from scratch is to normalize $\mathbf{x}_t$ by its variance to ensure it has unit variance before feeding it to the denoising network. This prevents performance issues caused by variance changes in $\mathbf{x}_t$ when $\mathbf{x}_0$ has the same mean and variance as $\epsilon$ [26]. However, in our image editing work using off-the-shelf diffusion models, we find that normalization does not improve content preservation or edit fidelity, as shown in Table 10.

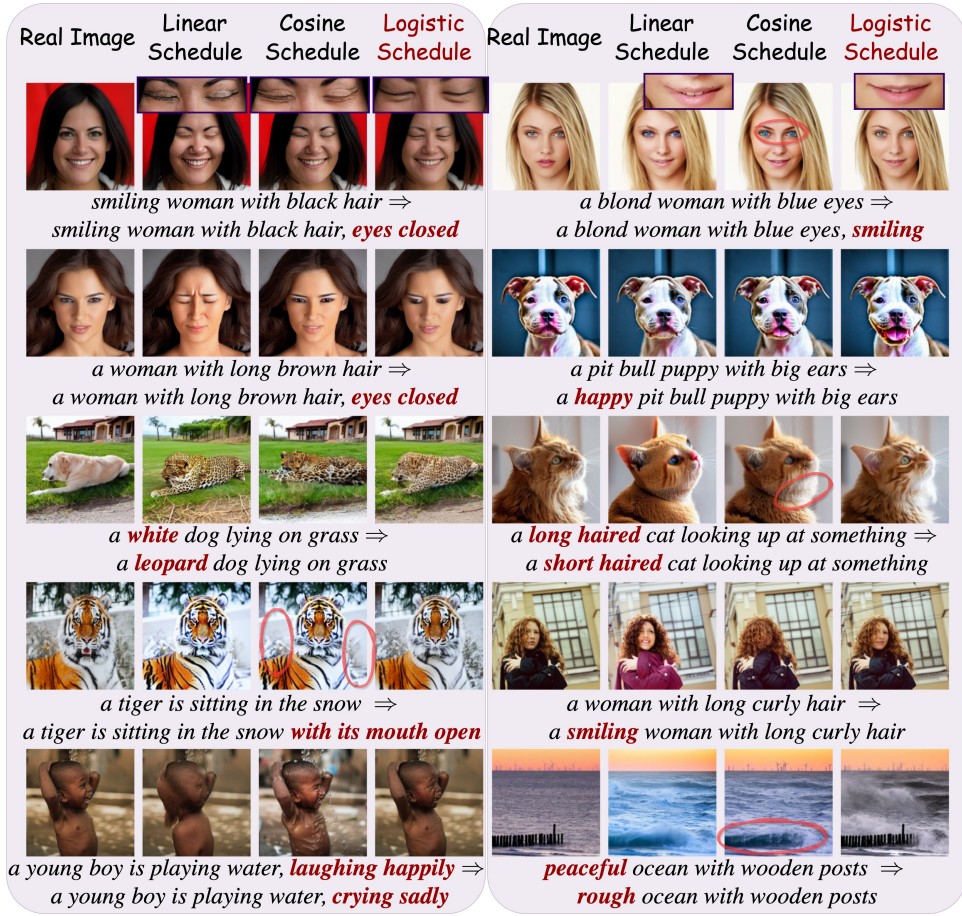

Figure 13: Comparison of linear, cosine, and logistic schedules in editing **attributes content**. The logistic schedule shows superior fidelity in preserving the original image content while accurately making the specified changes, without introducing artifacts or inconsistencies.

## F    Limitations and Future Works

This work endeavors to enhance inversion-based editing methods, focusing on improving noise schedule design during the inversion process. Even though this work reveals that modifying the noise schedule using off-the-shelf diffusion models can lead to editing improvements, there is still a lack of research on how other designs of noise schedules can lead to different effects. For example, is it possible to design dynamic adjustments of the noise schedule at each time step to achieve better results? Furthermore, the editing capabilities of the Logistic Schedule are inherently constrained by the limitations of inversion-based methods. For example, MasaCtrl [6] editing requires manual determination of timesteps and layers for attention control, limiting its ability to automatically adapt to diverse real-world objects with varying attributes.

Even though extensive experiments prove the effectiveness of the Logistic Schedule, it is worth diving deeper into the schedule's performance in the generation task. Due to computational resource constraints, we have not conducted training on the diffusion model from scratch to validate the full potential of the Logistic Schedule. Future work will include training diffusion models using the Logistic Schedule to validate its generation ability.

Another potential future research direction lies in exploring whether a steadier decrease in logSNR during perturbation, as described in Section 4, enhances editing quality. Additional experiments on both generation and editing are required to confirm if this trend extends to the generation process as well.

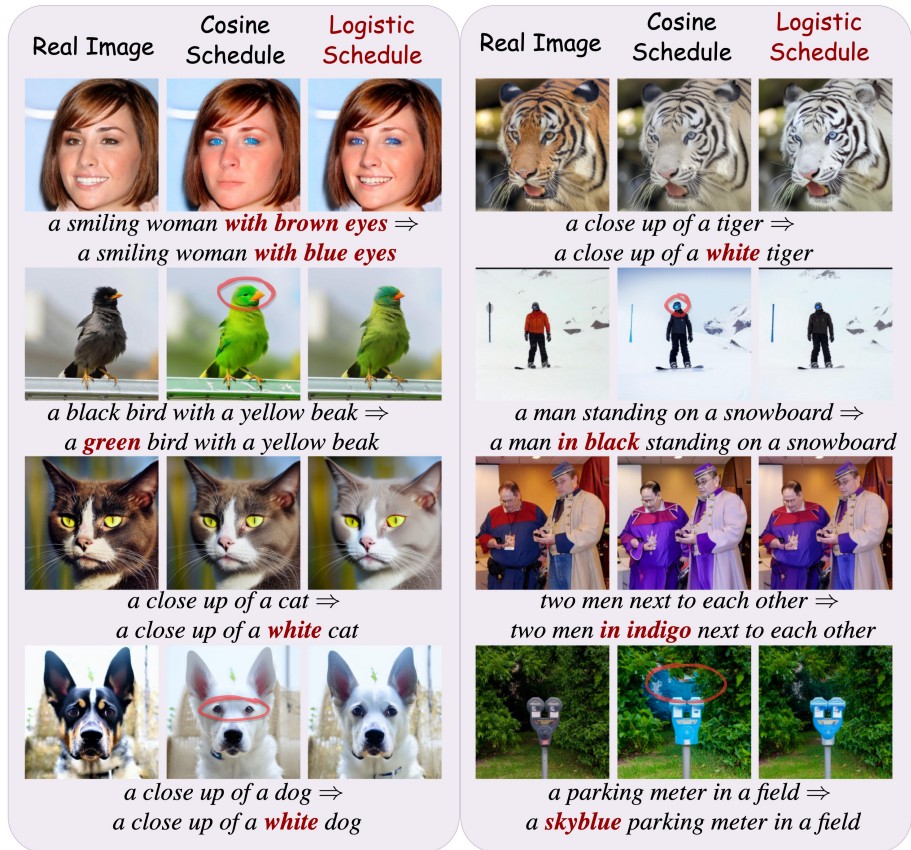

Figure 14: Comparison of linear, cosine, and logistic schedules in editing **color attributes**. The logistic schedule excels in maintaining natural and coherent lighting and shading, resulting in more realistic and seamless color changes without introducing visual inconsistencies.

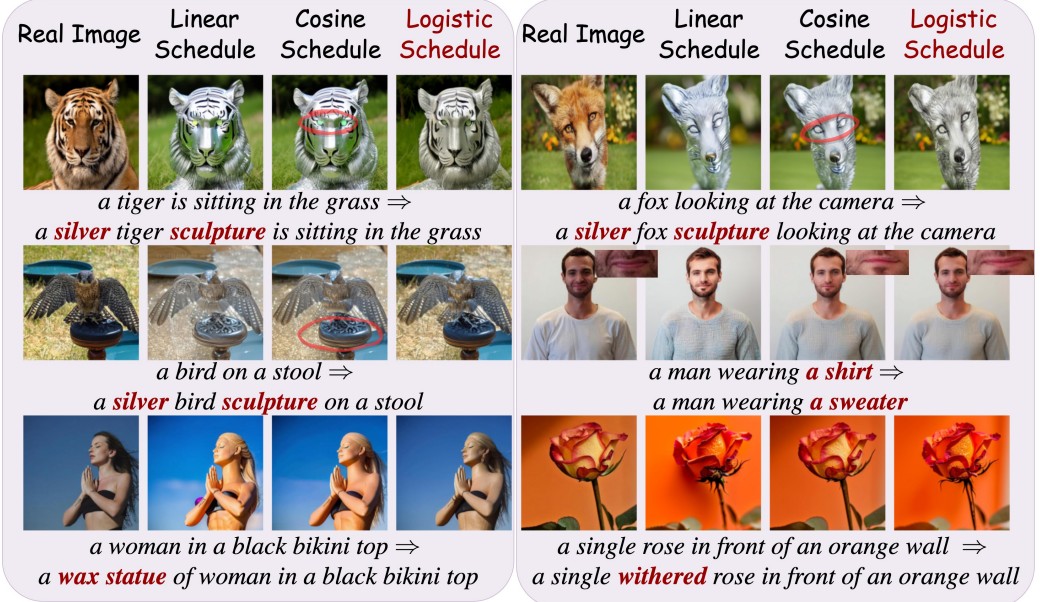

Figure 15: Comparison of linear, cosine, and logistic schedules in **changing material properties**. The logistic schedule excels in accurately rendering new material properties, such as reflections and textures while preserving the shape and form of the original objects, avoiding unrealistic artifacts, and ensuring a natural appearance.

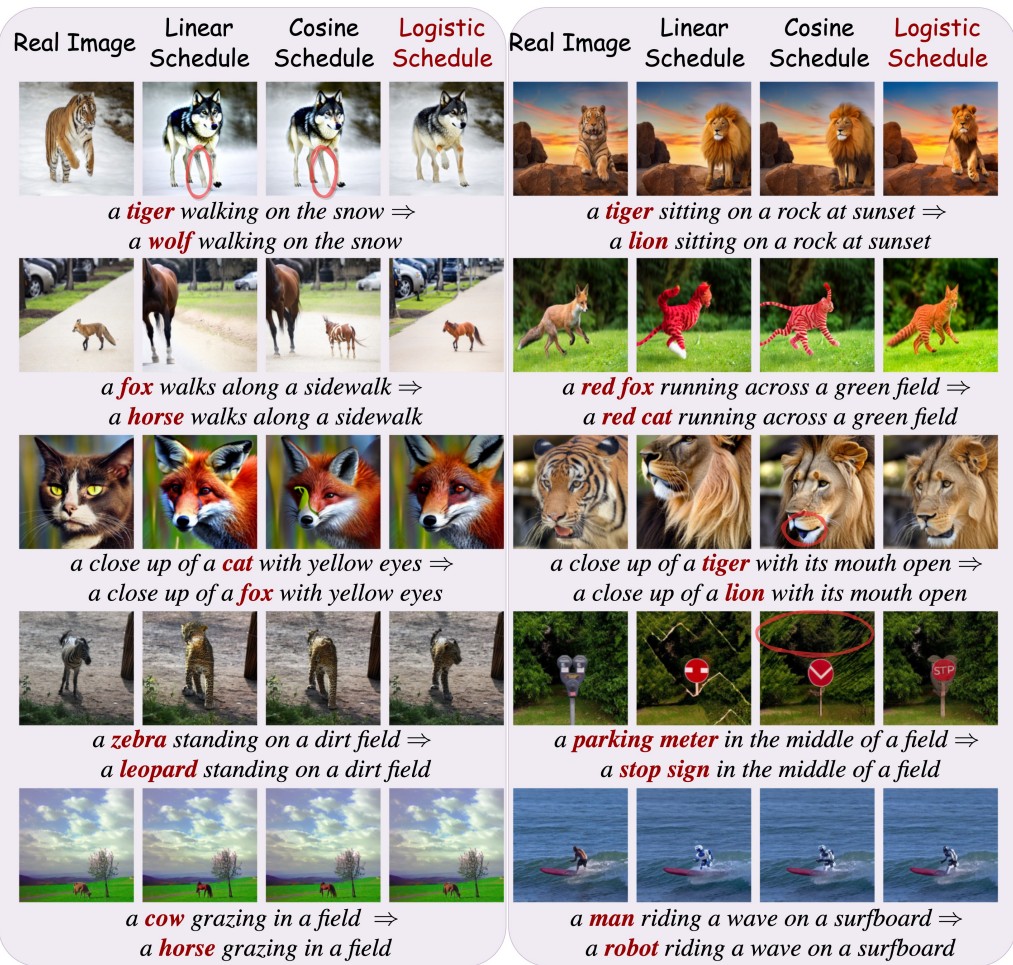

Figure 16: Comparison of linear, cosine, and logistic schedules in **switching objects**. The logistic schedule excels in maintaining the overall composition and context of the scene while seamlessly integrating the new objects with consistent lighting, shadows, and spatial relationships, ensuring a natural and coherent appearance.

# G  Broader Impacts

Our work introduces a novel editing technique for manipulating real images using state-of-the-art text-to-image diffusion models. While this technology could potentially be exploited by malicious parties to create fake content and spread disinformation, this is a common issue across all image editing techniques. Significant progress is already being made in identifying and preventing such malicious editing. Our research contributes to this effort by providing a detailed analysis of the inversion and editing processes in text-to-image diffusion models, thereby aiding in the development of more robust detection and prevention methods.

# H  Ethics Statement

Generative models for synthesizing images carry several ethical concerns, particularly when used by bad actors to generate disinformation or potentially displace creative workers through automation. These models, trained on large amounts of user data from the internet without explicit consent, may generate augmentations that resemble or copy such data. This issue is not unique to our work but inherent to large-scale models like Stable Diffusion, which we employ in our data augmentation strategy. To mitigate this, we allow for the deletion of harmful or copyrighted concepts from the model's weights before augmentation, ensuring such material cannot be copied during the process.

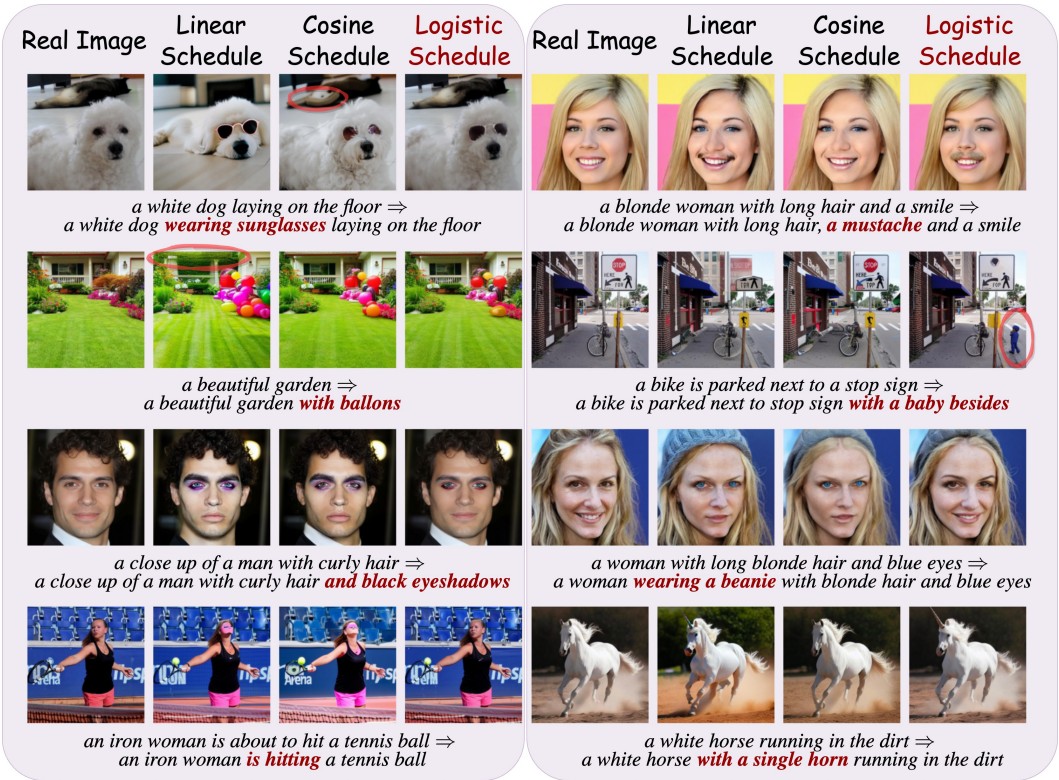

Figure 17: Comparison of linear, cosine, and logistic schedules in **adding objects**. The logistic schedule excels in naturally integrating the new objects into the scene, ensuring consistent lighting, shadows, and perspective with the existing elements, resulting in a realistic and seamless appearance.

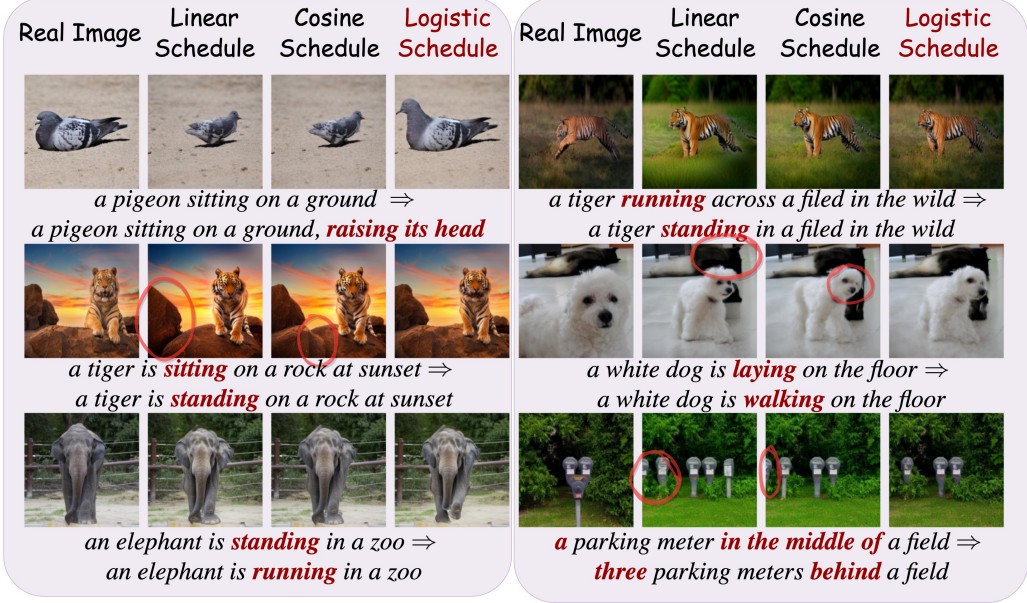

Figure 18: Comparison of linear, cosine, and logistic schedules in making **non-rigid** modifications. The logistic schedule excels in preserving anatomical correctness and natural appearance while making significant pose changes, ensuring smooth transitions and avoiding unnatural distortions.

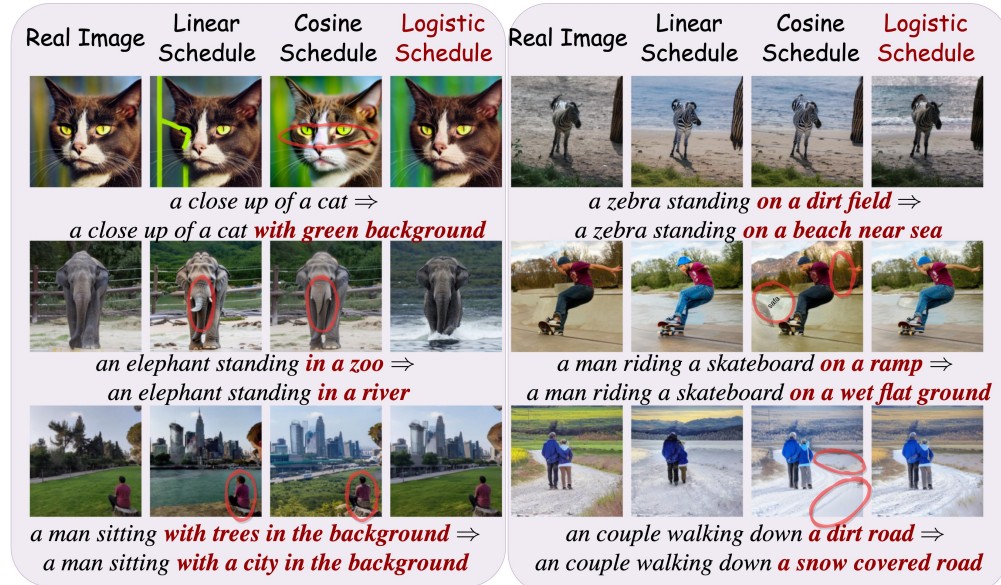

Figure 19: Comparison of linear, cosine, and logistic schedules in **transferring scenes**. The logistic schedule excels in seamlessly blending the new background with the foreground objects, maintaining consistent lighting, shadows, and color tones, avoiding visible seams, and ensuring a natural and coherent appearance.

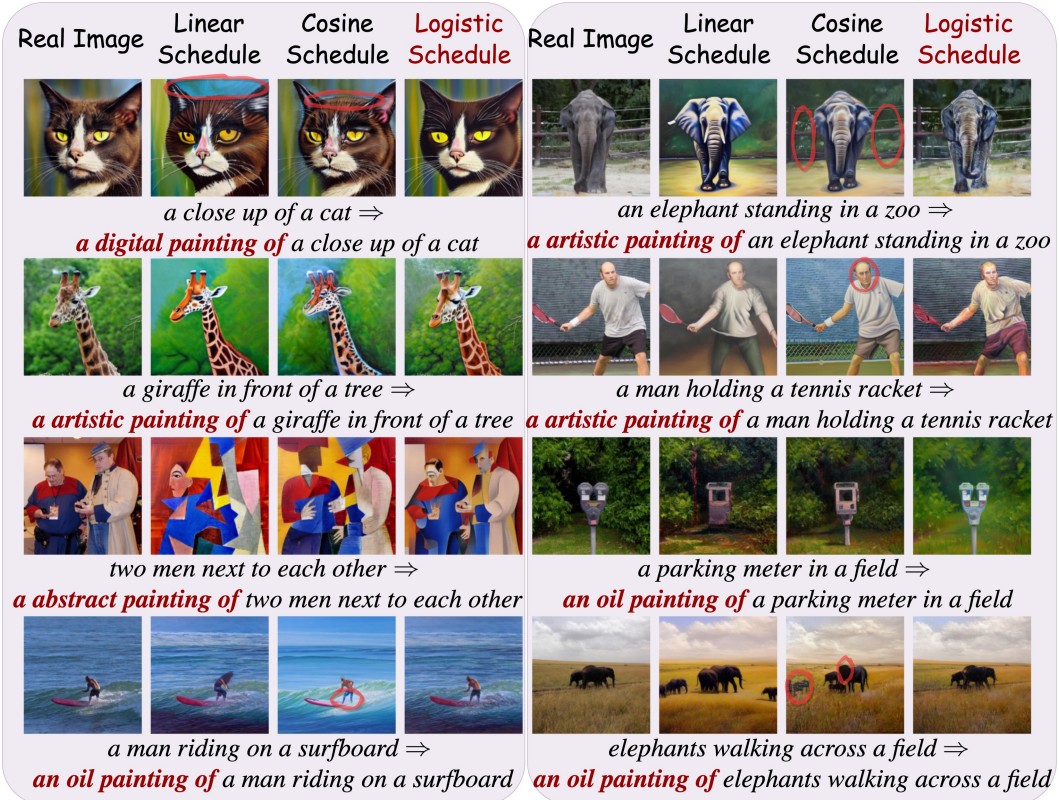

Figure 20: Comparison of linear, cosine, and logistic schedules in **transferring styles**. The logistic schedule excels in preserving essential details and content of the original image while accurately applying the new artistic style, ensuring consistency across the entire image and avoiding artifacts, resulting in a more natural and coherent style transfer.

Despite these concerns, these tools may also foster growth and improve accessibility in the creative industry.

