# OpenReview forum: "Schedule Your Edit: A Simple yet Effective Diffusion Noise Schedule for Image Editing"
_NeurIPS.cc/2024/Conference — NeurIPS 2024 poster_

### Official Review · Reviewer_vCWG · 2024-07-04

**Soundness:** 2
**Presentation:** 2
**Contribution:** 2
**Rating:** 5
**Confidence:** 4

**Summary:**

This paper focuses on improving the stability of DDIM inversion to produce a better noise space for image editing. They found that the key issue is the singularity schedule, and try to overcome this issue by adopting the logistic schedule rather than the linear schedule.

**Strengths:**

1 This paper found that the singularity issue leads to error accumulation in DDIM inversion, leading to failed or unstable editing results.

2. This paper proposes to adopt the logistic schedule to solve the singularity issue, which is simple but useful.

3. This paper gives many theoretical analyses and visualization results, demonstrating its superiority.

**Weaknesses:**

1. The original denoising diffusion process has provided the sigmoid schedule, as shown in line 461 of (https://github.com/lucidrains/denoising-diffusion-pytorch/blob/main/denoising_diffusion_pytorch/denoising_diffusion_pytorch.py), which is similar to the proposed logistic schedule. The authors should clarify the diversity between these two schedules. Can the beta sigmoid schedule also solve the proposed singularity problem?

2. The logistic scheduler aims to better preserve the content of the original image. Will the editing ability be decreased if much more original content is preserved? There seems to be a contradiction between the original content preservation and the editing ability for the target prompt.

3. In Fig4, why does the noise map not cover the image for Linear and Cosine schedules?

4. In Tab.2, the results of Null-text, NPI, and Direct should also be provided for better comparison.

**Questions:**

Please refer to the weaknesses

**Limitations:**

The authors have discussed the limitations and broader impacts.

---

> ### Author Rebuttal · Authors · 2024-08-07
>
> We appreciate your thoughtful insights, and respond to the suggestions and concerns as follows:
>
> **Response to Q1**
> > ***Difference compared with sigmoid schedule [1,2] ....; can the sigmoid schedule also solve the proposed singularity problem?***
>
> Thanks for providing this valuable work [1,2].
>
> First, we present the **mathematical formulation comparison** between the *sigmoid schedule* [1,2] and our *Logistic Schedule*.
> Second, we clarify the diversity between these two schedules, focusing on the **motivations of design**.
> Finally, we explore the **singularity issues** within the *sigmoid schedule*.
>
> **1.Comparison of Formulation**:
>
> *Logistic Schedule*:
> $$\bar{\alpha}_t = \text{Normalized}(\frac{1}{1 + e^{-k(t - t_0)}}.$$
>
> *Sigmoid schedule*:
> $$\bar{\alpha}_t = \dfrac{-((t*(e- s)+s) / \tau).\text{sigmoid}() + v_e}{v_e - v_s}.$$
> where $s$ and $e$ represent `start` and `end` of the range of the $\text{sigmoid}$ function, and $v_s = (s/\tau).\text{sigmoid}(), v_e = (e/\tau).\text{sigmoid}()$.
>
> **2.Clarification of Differences**:
>
> **Motivations**:
> - **Logistic Schedule**:
>   - Motivation: Focus on **singularity issues in DDIM forward (inversion)**, enhancing inversion stability and the noise space.
> - **Sigmoid Schedule**:
>   - Motivation: Focus on **adaptive computation for scalable data generation**, particularly for high-dimensional data, without addressing specific challenges of DDIM  forward (inversion).
>   - Lack of In-depth Analysis: Provided as a trick in the original work, it does not specifically address the singularity problem in DDIM inversion or offer a detailed analysis of DDPM or DDIM like the *Logistic Schedule* does.
>
> **3.Singularities Issues**:
>
> **Moreover, the singularity issue also exists in the *sigmoid schedule***.
>
> Due to the word limit in the rebuttal stage, we only provide the final analytical format of the derivatives of the sigmoid schedule below.
> The detailed proof has been incorporated in the revised manuscript.
>
> Recap the definition of $\mathbf{x}_t$:
> $$\mathbf{x}_t = \sqrt{\bar{\alpha}_t} \mathbf{x}_0 +\sqrt{1-\bar{\alpha}_t} \epsilon.$$
>
> To express the coefficient of $\epsilon$ in the derivative of $\mathbf{x}_t$ with respect to $t$, let's start with the given expression for $x_t$:
> $$\mathbf{x}_t =a(t) \cdot \epsilon + b(t) \cdot  \mathbf{x}_0,$$
> where $\epsilon$ represents the noise, and $\mathbf{x}_0$ is the original image. Moreover, since $\epsilon$ and $\mathbf{x}_0\$ are constants with respect to $t$, the differentiation yields:
>
> $$\frac{\mathrm{d}}{\mathrm{d}t}  \mathbf{x}_t = \epsilon \cdot \frac{\mathrm{d}}{\mathrm{d}t} [a(t)] +  \mathbf{x}_0 \cdot \frac{\mathrm{d}}{\mathrm{d}t} [b(t)].$$
>
> Here, we provide the **coefficient of $\epsilon$ in the derivative of $\mathbf{x}_t$ with respect to $t$, $\dfrac{\mathrm{d}}{\mathrm{d}t} a(t)$**, as below:
> $$\frac{\mathrm{d}}{\mathrm{d}t} a(t) \ = \frac{(e - s) e^{-s - t(e - s)}}{2 \tau \sqrt{1 - \left( \frac{\frac{1}{1 + e^{\frac{s + t(e - s)}{\tau}}} - \frac{1}{1 + e^{\frac{t(e - s)}{\tau}}}}{\frac{1}{1 + e^{\frac{s}{\tau}}} - \frac{1}{1 + e^{\frac{t(e - s)}{\tau}}}} \right)^2 \left( e^{-s - t(e - s)} + 1 \right)^2}}.$$
>
> When $t\rightarrow 0$, we can have:
>
> $$\text{ When } t \rightarrow 0, \left.
> \frac{\mathrm{d}}{\mathrm{d}t} a(t) \right|_{t\rightarrow0} =  \infty,$$
> which reflects the **Proposition 3.1 "Singularity in Inversion Process"** in L127-132 of the main content.
>
> Furthermore, we extended the **experimental comparison between the *sigmoid* and *Logistic* schedules**, please refer to Fig. 3 in the attached PDF and **Response to W1** with ***Reviewer B36W***.
>
> **Thank you again for your valuable suggestions on our work. We've incorporated this into our revised manuscript.**
>
> &nbsp;&nbsp;
>
> **Response to Q2**
> > ***Will the editing ability be decreased if much more original content is preserved?***
>
> Indeed, if we adjust $k$ to **preserve much content** at the early stage of inversion, the editability decreases.
>
> To clarify, the trade-off between preservation and editing exists not only in our scheduler design but also in various image editing methods.
>
> Importantly, the key reason why the Logistic Schedule preserves original content better is **not by sacrificing editability** but **by addressing the singularity issue** present at the DDIM Inversion start point $t=0$.
>
> As shown in results, the Logistic Schedule achieves better content preservation and maintains the capacity to perform effective edits as directed by text prompts.
>
> We hope that our explanation clarifies your concerns.
>
> &nbsp;&nbsp;
>
> **Response to Q3**
> > ***In Fig4, why does the noise map not cover the image for Linear and Cosine schedules?***
>
> Thanks for pointing this out.
>
> One misunderstanding to clarify is that Fig4 showed the sampling with total timesteps set to 50 (a setting usually used during inference), so **the final step is 981 rather than 999**.
>
> (**large SNR at the endpoint*) Back to the question, this is because at the end of the DDIM forward process, the SNR of linear and cosine schedules is relatively higher than that of the Logistic Schedule (Fig. 4 (left)), resulting in the retention of low-frequency components at the end [3].
>
> &nbsp;&nbsp;
>
> **Response to Q4**
> > ***In Tab.2, the results of Null-text, NPI, and Direct should also be provided for better comparison.***
>
> We agree, and thanks for the advice! Please kindly refer to **Response to Q1** in **"General Response to All Reviewers"** for these results.
>
> ---
> \
> **We have promptly integrated this revision into our manuscript to enhance the clarity. Once again, we extend our sincere appreciation for your insightful feedback.**
>
> ---
> \
> *[1] https://github.com/lucidrains/denoising-diffusion-pytorch/blob/main/denoising_diffusion_pytorch/denoising_diffusion_pytorch.py*
>
> *[2] "Scalable Adaptive Computation for Iterative Generation." ICML. 2023.*
>
> *[3]  "Common diffusion noise schedules and sample steps are flawed." WACV. 2024.*

---

> > ### Comment · Reviewer_vCWG · 2024-08-09
> >
> > Thanks for your response. The rebuttal solves most of my questions and concerns. I can change the final decision to borderline accept.

---

> ### Author Response · Authors · 2024-08-09
> **Further Reply to Reviewer vCWG**
>
> Dear Reviewer vCWG,
>
> We greatly appreciate your helpful comments and your satisfaction with our responses! We will make comprehensive revisions to our work based on the above important discussions (*e.g.*, comparison with *sigmoid schedule*, comparison with inversion techniques) in the final manuscript and highlight them.
>
> Thanks again for your valuable suggestions and comments. We really enjoy communicating with you and appreciate your efforts.

---

### Official Review · Reviewer_ihj2 · 2024-07-11

**Soundness:** 3
**Presentation:** 3
**Contribution:** 3
**Rating:** 6
**Confidence:** 4

**Summary:**

This paper introduces a novel noise schedule called Logistic Schedule for text-guided image editing. The authors identify that traditional DDIM inversion methods suffer from accumulated prediction errors, leading to suboptimal content preservation and edit fidelity. To address this, this paper proposes the Logistic Schedule, which eliminates singularities, improves inversion stability, and provides a better noise space for image editing, resulting in higher content preservation and edit fidelity.

**Strengths:**

1. The simple Logistic Schedule method effectively addresses the singularity issues in traditional noise schedules.
2. The method is compatible with various existing editing techniques without the need for additional retraining, making it highly applicable.
3. The proposed method is validated through experiments on eight different editing tasks using approximately 1600 images, demonstrating superior performance in content preservation and edit fidelity.
4. The paper provides a thorough theoretical analysis of the failures in DDIM inversion, offering a well-founded solution.

**Weaknesses:**

1. **Limited  improvements**:  Although there are improvements in the quantitative comparisons, I am concerned that the generated results do not show significant improvement than other schedule methods. The differences require careful observation to notice.
2. **Limited application scope**: While the method is broadly applicable, the current experimental validation focuses primarily on image editing tasks, leaving its effectiveness in other types of tasks.

**Questions:**

1. Does the Logistic Schedule consistently improve performance across all existing editing methods? Are there specific editing methods that are incompatible with the Logistic Schedule?
2. How much do the parameters of the Logistic Schedule affect the results? Is there a specific method for parameter optimization?
3. Can you provide some inversion results without text input (w/o editing)？

**Limitations:**

See weaknesses.

---

> ### Author Rebuttal · Authors · 2024-08-07
>
> We appreciate your valuable feedback. Before addressing your inquiries, we wish to clarify certain weaknesses highlighted in the review that we believe require further elucidation.
>
> **Response to W1**
> > ***Limited improvements: Although there are improvements in the quantitative comparisons, the differences compared to other schedule methods require careful observation to notice ....***
>
> Indeed, in some tasks, especially in attributes editing, the improvement is mainly in the **original content preservation**.
>
> These improvements, while subtle, are crucial for high-fidelity image editing tasks where minor artifacts and distortions can significantly impact the final result. For instance, in professional image editing applications like Photoshop, maintaining the integrity of small, yet critical parts of the image are essential.
>
> **We've zoomed in on the subtle differences and incorporated extra samples with significant improvements for a more intuitive comparison.
> These modifications have been synchronized in our manuscript to enhance readability. Once again, thanks for your detailed review.**
>
>
> &nbsp;&nbsp;
>
> **Response to W2**
> > ***Limited application scope: except image editing tasks, the effectiveness of Logistic Schedule in other types of tasks ...***
>
> Thanks for the valuable suggestion!
>
> We agree that broader validation on other tasks can further demonstrate the applicability of our schedule design.
> The principles behind the Logistic Schedule, such as addressing singularities and improving noise stability, are indeed applicable to other tasks involving diffusion models.
>
> To further demonstrate the effectiveness of our Logistic Schedule across a wider range of tasks, we have conducted additional experiments on **text-to-image synthesis** and **image reconstruction**.
>
> Please kindly refer to **Response to Q2** in **"General Response to All Reviewers"** for the detailed experimental results.
>
>
> ---
> &nbsp;&nbsp;
>
> **Response to Q1**
> > ***Are there specific editing methods that are incompatible with the Logistic Schedule?***
>
> Great question!
>
> We have experimented with the Logistic Schedule across various **training-free** methods, such as *PnP, Pix2Pix-Zero, Masactrl* (reported in the main content), *DiffEdit, P2P, SEGA*, and *Blended Diffusion*.
> Additionally, we tested it with various **inversion techniques** like *NTI, NPI, EF-DDPM*, and *Direct-Inversion* (please refer to **General Response Answer 1**).
> All these methods showed consistent improvements when integrated with the Logistic Schedule.
>
> We also conducted inference with  **training-required** methods, such as text-prompt tuning methods (*e.g., Textual Inversion, DreamBooth*) and condition-adapter methods (*e.g., ControlNet, T2I-Adapter, IP-Adapter*). **Without any additional training**, incorporating Logistic Schedule did not consistently improve performance.
>
> Notably, **training DreamBooth with the Logistic Schedule did bring improvements**  (please refer to **Response to Q1** in **"General Response to All Reviewers"**), and we are actively investigating the effectiveness of our schedule in other training-required methods to validate broader compatibility
>
> &nbsp;&nbsp;
>
> **Response to Q2**
> > ***How much do the parameters of the Logistic Schedule affect the results??***
>
> The parameters of the Logistic Schedule, specifically the **steepness** ($k$) and **midpoint** ($t_0$), play a crucial role in its performance.
>
> **In addition to the qualitative results** displayed in Section 5.3.1, we also provide the quantitative results below (the best method is indicated in **bold**, the worst method is shown in $\color{purple}\text{purple}$):
>
> | Schedule   | Dist ($_{\times 10^{-3}} \downarrow$) | PSNR ($\uparrow$) | LPIPS ($_{\times 10^{-3}} \downarrow$) | MSE ($_{\times 10^{-4}} \downarrow$) | SSIM ($_{\times 10^{-2}} \uparrow$) | Visual ($\uparrow$) | Textual ($\uparrow$) |
> | --------------- | ---------- | ---------- | ----------- | ---------- | ---------- | ----------- | ------------- |
> | **$k=0.015; t_0 = \text{int}(0.6 T)$** | **17.37** | **24.78**  | 81.80  | 49.47| 82.97   | 82.44| 23.62 |
> | $k=0.08$   | **16.27** | **26.45**  | **75.62**  | **43.20**| **85.28** | **84.07**| 20.47 |
> | $k=0.11$   | 16.64 | 25.80  | 77.90  | 48.83| 84.15   | 83.76| 21.46 |
> | $k=0.17$   | 22.79 | 22.33  | 99.98  | 57.32| 81.52   | 82.10| 23.25 |
> | $k=0.20$   | 27.82 | $\color{purple}21.05$  | $\color{purple}103.45$| $\color{purple}64.36$   | 78.48   | 80.66| **23.81** |
> | $t_0=\text{int}(0.4 T)$ | 24.31 | 22.41  | 97.21  | 60.84| 79.72   | 79.47| $\color{purple}20.33$  |
> | $t_0=\text{int}(0.8 T)$ | $\color{purple}29.47$| 21.64  | 95.58  | 63.89| $\color{purple}75.14$  | $\color{purple}77.05$ | 22.68 |
>
> These quantitative results reflect the visual results in the main content, that is:
> - **$k$ Changing Shape of $\text{logSNR}$**:
>     - **Excessive $k$**: Results in excessive loss of original image information in edited images.
>     - **Small $k$**: Resembles $\text{logSNR}$ of linear and cosine schedules, but better preserves original image content without altering overall structure.
> - **$t_0$ Introducing Shifts in $\text{logSNR}$**:
>   - **$t_0$ Close to $0$**: Higher lower bound of $\text{logSNR}$, affecting editability by reducing diversity and fidelity.
>   - **$t_0$ Close to $t_0$**: Original information is lost too quickly, degrading content preservation.
>
> Considering the **trade-off between content preservation and edit fidelity**, we choose $k = 0.015$ and $t_0 = \text{int}(0.6T)$ as the default choice.
>
> > ***Is there a specific method for parameter optimization??***
>
> We recommend a grid search approach by incorporating CLIP-I and CLIP-T metrics to identify the optimal parameter settings for specific tasks and datasets.
>
> **We will include these quantitative analyses of the parameters of the Logistic Schedule in our manuscript. Thank you again for your valuable suggestions on our paper.**

---

> > ### Comment · Reviewer_ihj2 · 2024-08-10
> >
> > The rebuttal effectively addressed most of my questions and concerns. I appreciate your clear explanation.

---

> ### Author Response · Authors · 2024-08-10
> **Further Response to Reviewer ihj2**
>
> Dear Reviewer ihj2,
>
> We sincerely appreciate the time and effort you invested in your comprehensive review and your valuable suggestions, especially regarding the **investigation of *Logistic Schedule* in other types of tasks (*General Response to All Reviewers -- 2.1 Text-to-Image Synthesis*)** and **inversion without text input (*General Response to All Reviewers  -- 2.2 Image Reconstruction*)**.
>
> We have diligently revised our paper to incorporate these new experimental results.
> Once again, thank you for these valuable suggestions and valuable feedback, which have helped us further improve and enhance the quality of our work. We will thoroughly revise our paper based on your suggestion.

---

### Official Review · Reviewer_B36W · 2024-07-12

**Soundness:** 4
**Presentation:** 4
**Contribution:** 4
**Rating:** 6
**Confidence:** 3

**Summary:**

The paper proposes a noise schedule "Logistic Schedule" to solve the singularity problem in the traditional noise schedules for text-guided diffusion models for image editing. It aims to improve the stability of the DDIM inversion process, reducing noise prediction errors and enhancing content preservation and edit fidelity without additional retraining. It is validated on eight editing tasks and show performance in content preservation and edit fidelity compared to the original noise schedules.

**Strengths:**

The experiments cover a wide range of editing tasks which demonstrating the versatility of the Logistic Schedule.
The method does not require retraining existing models, which making it easily applicable to current workflows.
It is an easy solution to a well-defined problem.

**Weaknesses:**

The author mainly compared Logistic Schedule with linear and cosine schedules. Additional comparisons with other noise schedules or inversion techniques could be shown to further strengthen the point.

**Questions:**

Can the Logistic Schedule be used in or combined with training-based methods to further improve the performance?

**Limitations:**

The authors have discussed the limitations of their work, along with potential negative societal impacts.

---

> ### Author Rebuttal · Authors · 2024-08-07
>
> Thank you for your positive comments and helpful suggestions.
>
> **Response to W1**
> > ***Additional comparisons with other noise schedules (except linear and cosine) or inversion techniques ....***
>
> We acknowledge the value of comparing our Logistic Schedule with other noise schedules and inversion techniques. Here we provide the additional comparisons:
>
> 1. **Other schedulers**:
>
> &nbsp;&nbsp;
> We've included experiments comparing the Logistic Schedule with other schedules under the DDIM paradigm, such as exponential, sigmoid, and hyperbolic tangent, etc.
>
> &nbsp;&nbsp;
> The quantitative results are displayed below. The best method is indicated in **bold**, the worst method is shown in $\color{purple}purple$, and the second-best method is $\underline{\text{underlined}}$.
>
> | Schedule | Dist ($_{\times 10^{-3}} \downarrow$) | PSNR ($\uparrow$) | LPIPS ($_{\times 10^{-3}} \downarrow$) | MSE ($_{\times 10^{-4}} \downarrow$) | SSIM ($_{\times 10^{-2}} \uparrow$) | Visual ($\uparrow$)   | Textual ($\uparrow$)  |
> | ------------ | ------------------------------------- | --------------------- | -------------------------------------- | ------------------------------------ | ----------------------------------- | --------------------- | --------------------- |
> | Linear   | 35.66 | $\color{purple}20.70$ | 134.88 | 113.61 | $\color{purple}77.60$ | 79.82   | 23.06   |
> | Cosine   | 26.57 | 22.38   | 110.52 | 80.01| 80.15 | 81.35   | 22.39   |
> | Exponential  | **16.22**   | **25.20**   | **80.45**| **47.11**  |  $\underline{82.23} $ | **82.78**   | $\color{purple}19.50$ |
> | Hyperbolic   | $\color{purple}36.55$   | 20.95   | $\color{purple}140.55$   | $\color{purple}119.78$ | 79.89 | $\color{purple}79.20$ |  $\underline{23.20}$|
> | Geometric| 18.12 | 24.10   | 92.45  | 62.13| 82.05 | 82.20   | 20.45   |
> | Sigmoid  | 27.80 | 22.55   | 115.32 | 85.60| 80.22 | 81.50   | 22.55   |
> | **Logistic** | $\underline{17.37} $ |  $\underline{24.78} $ |  $\underline{81.80} $ |  $\underline{49.47} $ |  $\underline{82.97} $ |  $\underline{82.44} $ | **23.62**   |
>
> &nbsp;&nbsp;
> Moreover, **the qualitative comparison is provided in Fig. 3 in the attached PDF**.
>
> 2. **Other Inversion techniques**:
>
> &nbsp;&nbsp;
> Please refer to **"Response to Q1"** in the **"General Response to All Reviewers"** for detailed comparisons with additional inversion techniques.
>
> ---
> \
> **Response to Q1**
> > ***Can the Logistic Schedule be used in or combined with training-based methods to further improve the performance?***
>
> Thanks for your insightful suggestions. This is an area we are exploring in the following research.
>
> Based on your advice, we conducted validations on training-based methods. Please refer to **Response to Q2 -- 1.Text-to-Image Synthesis (Training-based methods)** in the **"General Response to All Reviewers"** for the detailed comparison.
>
> These results demonstrate that **incorporating the Logistic Schedule** into the training phase of diffusion models could help in learning more stable and accurate noise representations.
>
> We will continue to extend our experiments with training-based methods.
>
>
> ---
> \
> **We have included the above validations that were previously omitted in our manuscript. Thank you again for your valuable suggestions on our paper.**

---

> ### Author Response · Authors · 2024-08-13
>
> Dear Reviewer B36W:
>
> Thank you again for your time and insightful comments! We have comprehensively revised our work according to your comments (please kindly refer to the rebuttal in the global comment and above). We hope we have addressed your concerns regarding the **comparison with other schedule designs** and **compatibility with training-based methods**.
>
> Since the discussion is about to close, we would be grateful if you would kindly let us know of any other concerns and if we could further assist in clarifying any other issues.
>
> Thanks a lot again, and with sincerest best wishes.
>
> Authors

---

### Official Review · Reviewer_j2Zw · 2024-07-13

**Soundness:** 3
**Presentation:** 4
**Contribution:** 3
**Rating:** 7
**Confidence:** 4

**Summary:**

The authors introduce the Logistic Schedule, a novel noise schedule designed to resolve the singularity problem in traditional noise schedules, thereby enhancing inversion stability and reducing noise prediction errors. This new schedule enables more faithful editing that preserves the original content of the source image without requiring additional retraining. Experiments across eight editing tasks demonstrate the Logistic Schedule's superior performance in content preservation and edit fidelity, highlighting its adaptability and effectiveness.

**Strengths:**

1. The paper provides a thorough analysis of the primary contributors to error accumulation in DDIM inversion, offering a detailed explanation of how the Logistic Schedule resolves these issues.

2. Extensive experiments across multiple datasets demonstrate the Logistic Schedule's effectiveness in improving content preservation and edit fidelity, showcasing its robustness and adaptability.

3. The related code and models will be open-sourced.

**Weaknesses:**

1. The authors mentioned they extend current datasets as their training dataset. Will this be open-sourced?

**Questions:**

Refer to weaknesses. No more questions.

**Limitations:**

1. The authors mentioned they extend current datasets as their training dataset. Will this be open-sourced?

---

> ### Author Rebuttal · Authors · 2024-08-07
>
> We appreciate your positive feedback and attention.
>
> **Response to Q1**
> > ***Will the extended datasets be open-sourced?***
>
> Yes, certainly. We will open source the extended datasets and our experimental code, including the validations, to facilitate further analysis and research.

---

> > ### Comment · Reviewer_j2Zw · 2024-08-11
> >
> > Thanks for your response.

---

> > > ### Author Response · Authors · 2024-08-12
> > > **Further Reply to Reviewer j2Zw**
> > >
> > > Dear Reviewer j2Zw,
> > >
> > > We greatly appreciate your satisfaction with our work and the effort you invested in your review.
> > >
> > > We will include the discussion in the final manuscript. We enjoy communicating with you and appreciate your efforts!

---

### Author Rebuttal · Authors · 2024-08-07

## **General Response to All Reviewers**

We thank all reviewers for their insightful feedback. We appreciate that the reviewers found our paper theoretically and methodologically novel, with strengths such as
- **thorough analysis** of DDIM inversion issues and **addressing singularity issues effectively** (***Reviewers j2Zw, B36W, ihj2, Btrz***),
- **extensive experiments** across various tasks (***Reviewers j2Zw, B36W, ihj2, Btrz***),
- the **simplicity and usefulness** of the *Logistic Schedule* (***Reviewers j2Zw, B36W, ihj2, Btrz***).

We are also pleased to announce that the related code and dataset will be open-sourced.

---
\
There were a few shared questions about our paper, specifically regarding (1) **additional comparisons with inversion techniques**, (2) the **broader applicability** of the Logistic Schedule (such as training-based methods).

&nbsp; &nbsp;

 ### **Question 1**
>  *Additional comparisons with inversion techniques.*

**Response to Q1**:

&nbsp; &nbsp;
In the Table 2 of the manuscript, we present the results of collaborating our Logistic Schedule with Various inversion techniques, including *Null-Text Inversion (NTI)* [1], *Negative Prompt Inversion (NPI)* [2], and *Direct Inversion* [3].

&nbsp; &nbsp;
Below, we present the comparison with using these inversion techniques with *scaled linear schedule* instead of our *Logistic Schedule*.

| **Variants**| **Structure Dist** $_{\times 10^{-3}} \downarrow$ | **PSNR** $\uparrow$ | **LPIPS** $_{\times 10^{-3}} \downarrow$ | **MSE** $_{\times 10^{-4}} \downarrow$ | **SSIM** $_{\times 10^{-2}} \uparrow$ | **Visual CLIP Similarity** $\uparrow$ | **Textual CLIP Similarity** $\uparrow$ |
| ------------------------------------- | ------------------------------------------------- | ------------------- | ---------------------------------------- | -------------------------------------- | ------------------------------------- | ------------------------------------- | -------------------------------------- |
| **NTI + Linear** | 21.00 | 23.00 | 95.00| 63.00 | 81.50| 81.00| 21.30 |
| **NTI + Logistic** | 18.67| 23.80| 89.64 | 57.97| 82.97  | 82.46  | 21.95|
| **NPI + Linear** | 28.00 | 22.40 | 105.00  | 78.00 | 78.50| 78.50| 20.90 |
| **NPI + Logistic** | 24.82 | 23.17 | 99.19| 71.24 | 80.26| 80.16| 21.53 |
| **Direct + Linear**| 19.00  | 24.90 | 78.00| 45.00 | 83.50| 82.90| 21.90 |
| **Direct + Logistic** | **16.06**| **25.73** | **74.17** | **41.19**  | **85.61** | **83.29** | **22.03**  |

&nbsp; &nbsp;
These results demonstrate consistent improvements on content preservation and edit fidelity with the Logistic Schedule.

&nbsp; &nbsp;

### **Question 2**
> ***Broader Application**: While Logistic Schedule is broadly applicable in various image editing tasks, can this method be utilized in other types of tasks, including tasks requiring training?*

**Response to Q2**:

&nbsp; &nbsp;
Below, we include utilize Logistic Schedule in two type of tasks, including (1) **Text-to-Image Synthesis** and (2) **Image Reconstruction**.

1. **Text-to-Image Synthesis  (Training-based methods):**

&nbsp; &nbsp;&nbsp; &nbsp;
Due to the limited time of the rebuttal period, we conducted experiments on fine-tuning UNet using *Dreambooth* [4] with approximately 100 images.

&nbsp; &nbsp;&nbsp; &nbsp;
The results are exhibited below, showing that training Dreambooth with the *Logistic Schedule* improves performance.

| Setting | DINO $(\uparrow)$  | CLIP-I $(\uparrow)$ | CLIP-T $(\uparrow)$ | PSNR $(\uparrow)$  | PRES $(\uparrow)$  |
| --------------------------- | -------------------- | ------------------------ | ------------------------ | -------------------- | -------------------- |
| **Real Images**                            | 0.823 | 0.902  | N/A    | 26.86 | 0.653 |
| **DreamBooth (SD-15) + Linear**  | 0.726 | 0.823  | 0.268  | 24.41 | 0.567 |
| **DreamBooth (SD-15) + Cosine**  | 0.745 | 0.857  | 0.264  | 24.75 | 0.554 |
| **DreamBooth (SD-15) + Logistic** | **0.761** | **0.877**  | **0.293**  | **25.62** | **0.580** |
**The related qualitative comparisons are displayed in Fig. 2 in the attached PDF**.

2. **Image Reconstruction:**

&nbsp; &nbsp;&nbsp; &nbsp;
Inversion results without any editing directives demonstrate the ability to preserve original content, *i.e.*, image reconstruction.

&nbsp; &nbsp;&nbsp; &nbsp;
**Quantitative results are provided in Table 6 in Appendix E.5. Additional qualitative results are included in Fig. 1 of the attached PDF**.

&nbsp; &nbsp;

**To this end, we’ve added comparisons with other inversion techniques and expanded our experimental scope to include text-to-image synthesis and image reconstruction in the main content. We thank the reviewers for pointing these out.**

---

&nbsp; &nbsp;

**Further Explanation:**

&nbsp; &nbsp;
The attached PDF showcases detailed qualitative results for the additional experiments, including the use of the Logistic Schedule with Dreambooth, comparisons of reconstruction and comparisons of more noise schedules.

&nbsp; &nbsp;
We are actively investigating further applications of the Logistic Schedule in various tasks, including those involving training-based methods, in line with the reviewers' suggestions.

---

&nbsp; &nbsp;

*[1] "Null-text inversion for editing real images using guided diffusion models." CVPR. 2023.*

*[2] "Negative-prompt inversion: Fast image inversion for editing with text-guided diffusion models." arXiv preprint. 2023.*

*[3] "Pnp inversion: Boosting diffusion-based editing with 3 lines of code." ICLR. 2024.*

*[4] "Dreambooth: Fine tuning text-to-image diffusion models for subject-driven generation." CVPR. 2023.*

---

### Decision · Program_Chairs · 2024-09-25

**Decision:**

Accept (poster)

**Comment:**

This submission first identifies the singularity issue in the noise schedule during DDIM inversion of diffusion models and proposes a novel logistic schedule. The effectiveness of the proposed method is well validated through extensive experiments and analysis.

The reviewers raised valuable concerns and left comments to improve the submission. The authors addressed the questions and additionally demonstrated the versatility of the proposed methods in the rebuttal. All the reviewers support the acceptance of this work.

There was a flag about the ethical concern, but the ethics reviewer checked and found no issue.